# BrainMoE: Cognition Joint Embedding via Mixture-of-Expert Towards Robust Brain Foundation Model

**Ziquan Wei**  **Tingting Dan**  **Tianlong Chen**  **Guorong Wu**\*

Departments of Computer Science and Psychiatry
University of North Carolina at Chapel Hill
Chapel Hill, NC 27599
{ziquanw,tianlong}@cs.unc.edu;{Tingting_Dan,grwu}@med.unc.edu

## Abstract

Given the large scale of public functional Magnetic Resonance Imaging (fMRI), e.g., UK Biobank (UKB) and Human Connectome Projects (HCP), brain foundation models are emerging. Although the amount of samples under rich environmental variables is unprecedented, existing brain foundation models learn from fMRI derived from a narrow range of cognitive states stimulated by similar environments, causing the limited robustness demonstrated in various applications and datasets acquired with different pipelines and limited sample size. By capitalizing on the variety of cognitive status as subjects performing explicit tasks, we present the mixture of brain experts, namely BrainMoE, pre-training on tasking fMRI with rich behavioral tasks in addition to resting fMRI for a robust brain foundation model. Brain experts are designed to produce embeddings for different behavioral tasks related to cognition. Afterward, these cognition embeddings are mixed by a cognition adapter via cross-attention so that BrainMoE can handle orthogonal embeddings and be robust on those boutique downstream datasets. We have pre-trained two existing self-regressive architectures and one new supervised architecture as brain experts on 68,251 fMRI scans among UKB and HCP, containing 12 different cognitive states. Then, BrainMoE is evaluated on a variety of applications, including sex, age prediction, human behavior recognition, disease early diagnosis of Autism, Parkinson's disease, Alzheimer's disease, and Schizophrenia, and fMRI-EEG multimodal applications, where promising results in eight datasets from three different pipelines indicate great potential to facilitate current neuroimaging applications in clinical routines.

## 1 Introduction

Like foundation models for other topics, brain foundation models aim to learn feature representation fundamentally from large-scale data of neuroimaging. Functional Magnetic Resonance Imaging (fMRI) of the brain, as it offers insight into the relationship between functional fluctuations and human behavior [1], is critical to discovering the enigma of human cognition and promoting clinical applications. Blood-Oxygen-Level Dependent (BOLD) signal in fMRI measures neuronal activity. Such raw signals are preprocessed as timeseries of regional mean or the functional connectivity (FC) by coefficient correlation for analysis with high Signal-to-Noise Ratio (SNR) [6]. While the exploration of brain foundation model has expanded to various masking strategies for either (latent) BOLD or FC reconstruction [23, 11, 34], previous works formulating this problem as transferring

---

\*Corresponding author.

39th Conference on Neural Information Processing Systems (NeurIPS 2025).

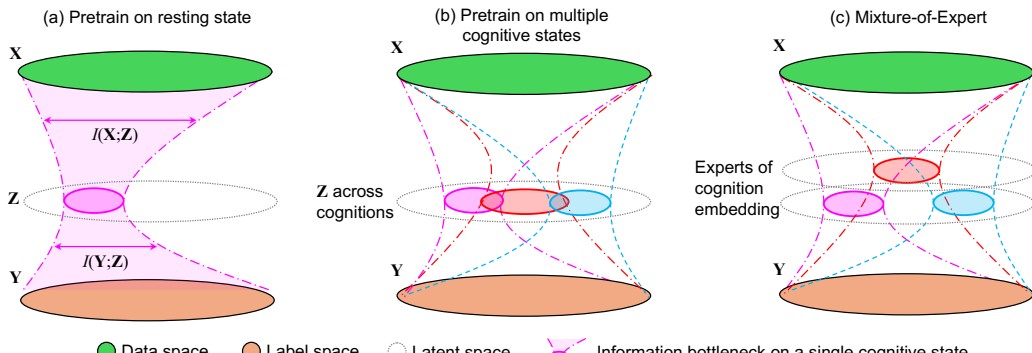

Figure 1: Motivation of BrainMoE through the lens of the information bottleneck theory. (a) Feature representation learning makes an information bottleneck between data and label space, where $\mathbf{X}$ denotes data, $\mathbf{Z}$ denotes latent feature representation, $\mathbf{Y}$ is the label of data, and $I(\cdot;\cdot)$ is the mutual information. (b) A model pre-trained on multiple cognitive states may compromise the underlying heterogeneity between different states, where $\mathbf{Z}$ cannot be optimal for all states. (c) The mixture of brain experts dedicated to diverse cognitive states leads to a joint cognition embedding so that the downstream applications can be advanced by stratified pre-training on rich behavioral tasks.

self-regressive methodologies from natural language and image to neuroimaging ignored the inter-correlation between non-imaging phenotypes [14]. Furthermore, they are restricted by a narrow range of one or two cognitive states, e.g., the resting state, causing samples with behaviors other than resting to be overlooked. Due to the lack of explicit designs to utilize the complete brain fMRI dataset with respect to neuroscience knowledge, a mixture of brain experts is proposed towards a robust brain foundation model in this work.

UK Biobank (UKB) [20] and two Human Connectome Projects (HCP) [29, 5] that contain healthy subjects 22 to 100 years of age are mainly used as pre-training datasets since they have a large scale. Previously, most subjects in resting state among UKB and HCP were included in BrainMass [34] and BrainJEPA [11]. While BrainLM [23] involved an additional tasking state in UKB, its performance has shown worse than others. Even though BrainMass has collected the most available published resting fMRI data on the OpenNeuro platform [22], ten available cognitive states in HCP datasets were ignored. It is intuitive to train a single model with all available data. However, as shown in Fig. 1 (a) and (b), simply pre-training with all cognitive states results in a single model being suboptimal to samples with different cognitive states, e.g., the red information bottleneck established by mutual information is suboptimal in the latent space where cognition related behaviors are variable. In fact, a single model is observed to compromise the underlying heterogeneity between cognitive states derived from diverse neural circuits stimulated by different behaviors [25]. This issue necessitates mixing experts specialized in different cognitive states, as shown in Fig. 1 (c), where each expert produces the cognition embedding, that is, a feature representation stratified by cognitive states.

On the other hand, tremendous efforts have been made to benchmark generally purposed models [8, 24, 10] and brain-dedicated architectures [19, 17, 4, 31] on brain fMRI data. A common observation on the results is that performance is diverse using BOLD or FC as the model input for different datasets, leading to related brain fMRI analysis works being categorized by types of input: (1) BOLD foundation models [23, 11] and (2) FC foundation models [15, 34]. Nevertheless, this reduces the adaptability of previous brain foundation models for datasets that fit better with a type of input differentiated from the pre-training stage. Mixture-of-experts (MoE) cooperating with router and adapter [35, 37] has demonstrated great potential for multimodal, referring to BOLD and FC in our data, and multitask, referring to multiple cognitive states. Therefore, a novel cognition adapter is proposed to facilitate BrainMoE as a robust brain foundation model learning from various cognitive states. Although adapters in MoE for language and vision fields are using small architecture like a multilayer perceptron (MLP) [18], a high scalability of the cognition adapter can ensure the transformation from cognition embeddings to objectives. Given that MLP is not scalable (see Appendix), a Transformer decoder is utilized for adapting BrainMoE to downstream applications.

To this end, this work has three contributions: (1) We propose BrainMoE, an MoE framework for brain fMRI data that towards high robustness for downstream tasks with different pipelines and limited sample size. (2) A cognition adapter is designed to adapt embeddings from experts pre-trained

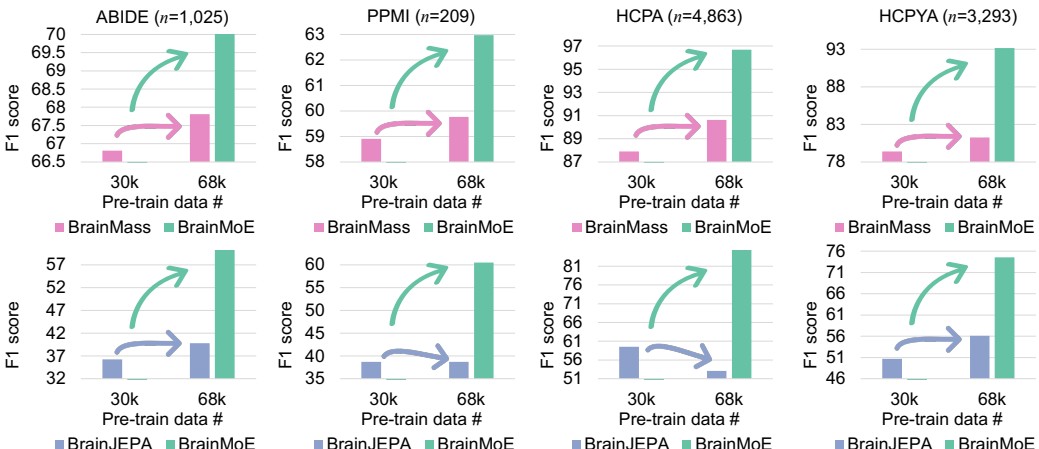

Figure 2: Increasing the scale of pre-training data of a brain foundation model leads to a marginal or negative performance boost due to 11 overlooked cognitive states. Four columns are four downstream applications, where ABIDE and PPMI are disease recognition, and HCPA and HCPYA are human behavior recognition. Two rows are two expert architectures, where BrainMass uses FC and BrainJEPA uses BOLD.

with various cognitive states, regardless of the input type, for finetuning. (3) Two existing brain foundation models pre-trained in self-supervised manners and one cognition classifier pre-trained in a supervised manner are both evaluated as experts in BrainMoE, where experts pre-train on 68,251 fMRI scans among UKB and HCPs and fine-tune on various applications, including sex prediction, human behavior recognition, and disease early diagnosis of Autism, Parkinson's disease, Alzheimer's disease, and Schizophrenia among seven datasets.

## 2 Preliminaries

**Brain foundation models**   To the best of our knowledge, BrainLM [23] represents the first brain foundation model. It applied Masked Autoencoding (MAE) to BOLD signals reconstruction. However, densely filling the entire fMRI time series can impair the model's capacity to differentiate between noise and meaningful signals. Prior work [3] has demonstrated that masked pretraining in generative frameworks such as MAE often yields suboptimal results in off-the-shelf evaluations, such as linear probing. Similarly, BrainJEPA [11] introduces an alternative architecture employing a distinct JEPA-based masking strategy, addressing BrainLM's limitations by drawing on insights from I-JEPA [3]. Although BrainJEPA reports superior performance relative to linear probing, it does not explicitly incorporate pre-training with tasking fMRI. BrainMass [34] used a larger pre-training dataset (see Appendix) and a matching objective between pseudo FC matrices as a novel framework. Whilst, it used sololy the resting fMRI and overlooked more than 38k tasking fMRI in the dataset.

**Resting- and tasking-state fMRI**   Neuroimaging data contains brain activity reflecting the interaction between functional fluctuations and human behavior. Studies controlled subjects in a resting or explicit tasking state during the data acquisition to offer distinct, complementary perspectives on brain function [36]. Large-scale studies [29, 5, 20] are observed to collect at least one tasking state in addition to the resting state, while brain foundation models mainly pre-train on the portion under the resting state, overlooking half or more data in the datasets. The intuitive method of mixing all data together may compromise the underlying heterogeneity between different states. Fig. 2 shows that the improvement gained from pre-training with 38k more data containing 11 overlooked cognitive states is marginal on four downstream classifications: ABIDE is 2-class classification for Autism diagnosis, PPMI is 4-class for staged Parkinson's disease, HCPA is 4-class, and HCPYA is 7-class for human behavior recognition. In contrast, BrainMoE can bring an impressive performance enhancement by stratifying and adapting cognitive states. Note that ABIDE and PPMI are preprocessed by the pipeline of [33], which is different from our pipeline (see Appendix) for HCPs and UKB datasets. Furthermore, BrainMass and BrainJEPA reconstruct FC and BOLD latent features, respectively.

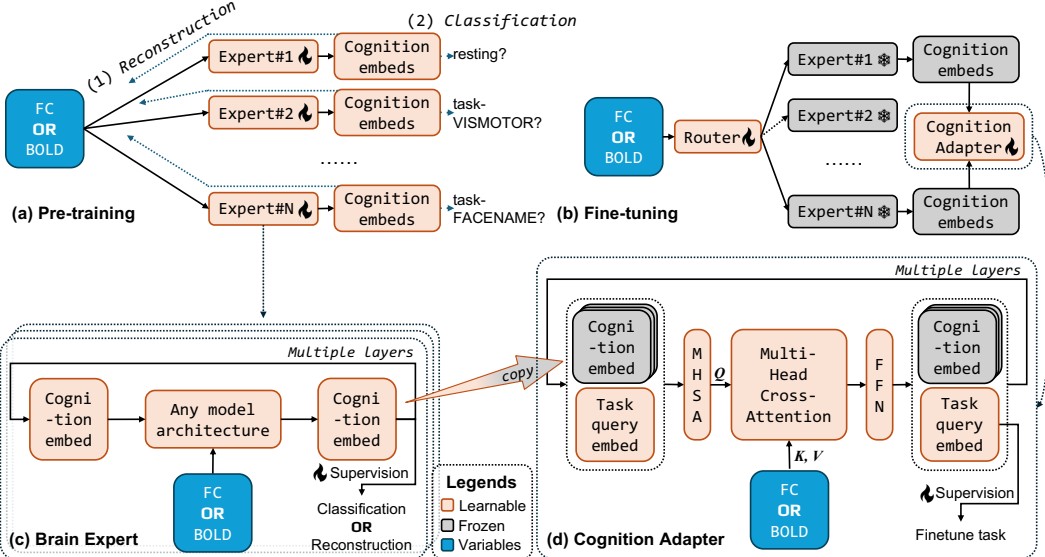

Figure 3: Framework of BrainMoE. **(a) Pre-training** has two options: (1) Previous models train with a reconstruction objective, where samples used for BrainMoE pre-training are stratified by cognitive states. (2) A new architecture of cognition classification can be set as the objective via cross-entropy between embeddings and cognitive states. **(b) Fine-tuning** a router for the expert selection and a cognition adapter for the combination of frozen experts. **(c) Brain expert** is adaptable to any model architecture, learning feature representation from either FC or BOLD signal, where the vector of latent feature before the final predictive layer is used as a cognition embedding. **(d) Cognition adapter** is a Transformer decoder, where MHSA stands for multi-head self-attention, and FFN is a feedforward network.

These primary results demonstrate empirical and clear evidence to support the motivation to stratify data according to cognitive states for multiple experts and learn joint cognition embeddings.

## 3 Methods

BrainMoE is designed to work with arbitrary brain foundation models as experts and to cooperate with a router and an adapter for fine-tuning on the downstream. Assume the input, BOLD or FC, is denoted by $\mathbf{X} \in \mathbb{R}^{M \times C_{in}}$ with $M$ regions of the brain atlas and $C_{in}$ channels of input vector. Target of brain experts is to produce cognition embeddings, $\mathbf{Z}$. For the router and adapter, it is to predict $\mathbf{Y}$ for downstream applications given pre-trained experts.

### 3.1 Framework

The framework of BrainMoE is separated into two stages as shown in Fig. 3 (a) pre-training and (b) fine-tuning, where $N$ experts, denoted by $f(\cdot) : \mathbb{R}^{M \times C_{in}} \to \mathbb{R}^{C_{hid}}$ with $C_{hid}$ the hidden channel number, learn from large-scale datasets containing subjects explicitly tasking on $N$ cognition-related behaviors to produce a variety of cognition embeddings, and fine-tune a router for expert weights $\mathbf{P} \in \mathbb{R}^{N}$, for selecting top-$k$ ($k \in [1, N]$) experts, and a cognition adapter for predicting downstream tasks based on cognition embeddings. Following the observed performance that relies on input type, preprocessing pipeline, and model architecture, BrainMoE has a framework suitable to experts with no requirement for data type and architecture.

### 3.2 Expert pre-training

In Fig. 3 (a), two objectives can be used to pre-train the brain expert, the reconstruction and the classification. (1) The pre-training of existing brain foundation models is reconstructing latent feature of input, FC or BOLD, from its masked version via a bottleneck or transformer encoder architecture. We utilized BrainMass or BrainJEPA as candidates of brain experts. The latent feature is produced by

existing architectures as the cognition embedding, denoted by $\mathbf{Z} \in \mathbb{R}^{C_{hid}}$. Each expert is pre-trained with the data that has the same cognitive state.

$$\mathbf{Z} := f(\mathbf{X}) = \arg\min_{\mathbf{Z}} ||\mathbf{Z} - g(\mathbf{X})||^2, \tag{1}$$

where $g$ is the target network. As shown in Fig. 3 (c), the brain expert can be any architecture that produces a latent feature representation, which is frozen and copied for the downstream. The pre-training objective is not restrict to the reconstruction or a new classification for cognitive states.

In Fig. 3 (a) (2), we propose a new pre-training objective, cognitive state classification, to explicitly learn from the cross-entropy between the latent feature and the cognitive state, $CELoss(\rho(\mathbf{Z}), \mathbf{Y}_{cog})$, where $\rho : \mathbb{R}^{C_{hid}} \to \mathbb{R}^1$ is a linear layer, and $\mathbf{Y}_{cog}$ is the binary label of a cognitive state. The architecture for this expert is the same as the cognition adapter introduced in the next section.

### 3.3 Cognition adapter fine-tuning

The architecture of the cognition adapter is designed as a Transformer decoder shown in Fig. 3 (d). The purpose of this adapter is to adapt multiple experts from a stratified feature representation based on cognitive states to a downstream application, a classification task in this work.

Assume that the token vectors shown in dashed rectangle in Fig. 3 (d) is denoted by $\bar{\mathbf{Z}} \in \mathbb{R}^{(k+P) \times C_{hid}}$, where $P$ is the class number in the downstream application. Note that $\bar{\mathbf{Z}}_{:k} := \mathbf{Z} \odot \mathbf{P}$ representing the top-$k$ cognition embeddings from experts and $\bar{\mathbf{Z}}_{k:(k+P)}$ denotes randomly initialized task query embeddings. It is also a cognition classifier without $\bar{\mathbf{Z}}_{:k}$. Then, as demonstrated in the architecture, a layer of the adapter starts at a multi-head self-attention (MHSA), $\bar{\mathbf{Z}} = \texttt{Softmax}(QK^T/\sqrt{C_{hid}})V$, with following definitions

$$Q := \bar{\mathbf{Z}}\bar{\boldsymbol{\alpha}}_h, K := \bar{\mathbf{Z}}\bar{\boldsymbol{\beta}}_h, V := \bar{\mathbf{Z}}\bar{\boldsymbol{\gamma}}_h, \tag{2}$$

where $\bar{\boldsymbol{\alpha}}_h, \bar{\boldsymbol{\beta}}_h, \bar{\boldsymbol{\gamma}}_h \in \mathbb{R}^{C_{hid} \times C_{hid}}$ are learnable parameters, and $h$ is the head index. Last, a multi-head cross-attention brings the information from the raw input to the task embeddings. Suppose $\mathbf{I} \in \mathbb{R}^{M \times M}$ is FC matrix. Cross-attention between $\bar{\mathbf{Z}}$ and $\mathbf{I}$ with alternative definitions

$$Q := \mathbf{I}\hat{\boldsymbol{\alpha}}_h, K := \bar{\mathbf{Z}}\hat{\boldsymbol{\beta}}_h, V := \mathbf{I}\hat{\boldsymbol{\gamma}}_h, \tag{3}$$

where $\hat{\boldsymbol{\alpha}}_h, \hat{\boldsymbol{\gamma}}_h \in \mathbb{R}^{M \times C_{hid}}, \hat{\boldsymbol{\beta}}_h \in \mathbb{R}^{C_{hid} \times C_{hid}}$ are learnable parameters. FFN denotes a feedforward network constructed by MLP. Note that the bias in linear layers is omitted in this section for clarity. Finally, after multiple layers of the cognition adapter, a linear layer, $\mathbb{R}^{P \times C_{hid}} \to \mathbb{R}^{P \times 1}$, takes only the task query and produces the logistic prediction to accomplish fine-tuning on a downstream application.

## 4 Experiments

We evaluate the proposed BrainMoE on 3 pre-training datasets, including UK Biobank (UKB), HCP Aging (HCPA), and HCP Young Adult (HCPYA), and 7 downstream datasets, including ADNI, ABIDE, PPMI, Taowu, SZ, HCPA, and HCPYA. UKB and two HCPs contain 68,251 scans of brain fMRI from 21,797 subjects under 12 different cognitive states on resting or tasking. Five disease-related datasets contain more than 1,500 subjects under the same resting state but various health status.

To comprehensively evaluate and showcase the performance, we conduct experiments on both randomly initialized and pre-trained models across tasks involving disease, sex, and brain state recognition. Specifically, our study aims to address two key research questions: (**RQ1**) To what extent does BrainMoE improve the prediction performance from the baseline using different expert architectures? (**RQ2**) Which pre-training objectives are the most robust across various downstream applications? Additionally, we provide ablation studies to further support our findings.

**Datasets** We preprocess fMRI and partition brain regions using the AAL atlas [28] for UKB, HCPA, HCPYA, and ADNI. Details can be found in the Appendix. Other datasets are preprocessed by [33]. SZ is an in-house data preprocessed by a third party.

**UK Biobank (HCPA)** dataset [20] is a large-scale dataset with MRI data. There are fMRI ($n$=51,780) involved in this work. It consists of one resting state and one tasking state that engages cognitive and sensory-motor [13].

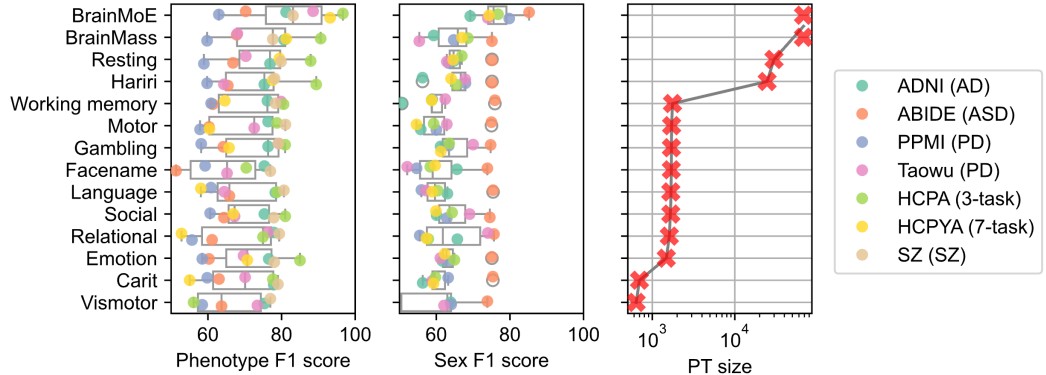

Figure 4: The performance of BrainMoE and BrainMass pre-trained with all samples, and $n$=12 pre-trained individual experts on phenotypic and sex classifications among 7 datasets, where scores lower than 50% are hidden for clarity, and the pre-training (PT) size ranges from 68,251 to 637.

**The Lifespan Human Connectome Project Aging (HCPA)** dataset [5] is instrumental in task recognition research, offering a comprehensive view of the aging process. It includes data from 717 subjects, encompassing fMRI records ($n$=4,863) with four human behaviors associated with memory, sensory-motor, and the resting state.

**The Human Connectome Project Young Adult (HCPYA)** dataset [29] has tackled key aspects of the neural pathways that underlie brain function and behavior via high-quality neuroimaging data in over 1,100 healthy young adults. It includes data from seven human behaviors associated with various cognitive tasks, e.g., language and working memory.

**Alzheimer's Disease Neuroimaging Initiative (ADNI)** dataset [32] serves as an invaluable resource, featuring a collection of pre-processed fMRI ($n$=138) and including clinical diagnostic labels. It encompasses a spectrum of cognitive states: Cognitive Normal (CN), Subjective Memory Complaints (SMC), Early-Stage Mild Cognitive Impairment (EMCI), Late-Stage Mild Cognitive Impairment (LMCI), and Alzheimer's Disease (AD). Considering the class unbalance issue, we simplified these categories into two broad groups based on disease severity: we combined CN, SMC, and EMCI into 'CN' group, while LMCI and AD were grouped as the 'AD' group.

**Parkinson's Progression Markers Initiative (PPMI)** dataset [33] presents a substantial collection of data from 209 subjects. It encompasses states of mental health: normal control, scans without evidence of dopaminergic deficit (SWEDD), prodromal, and Parkinson's disease (PD).

**Taowu** [33] is one of the earliest image datasets released for Parkinson's and contains 40 subjects.

**Autism Brain Imaging Data Exchange (ABIDE)** dataset [33] presents data from 1,025 young adults. The initiative aggregated fMRI data collected from laboratories around the world to support the research on Autism Spectrum Disorder (ASD).

**Schizophrenia (SZ)** is the in-house data that contains 189 subjects. There are 30 converted and 159 nonconverted.

**Implementation**   Following previous works, our experiments are conducted with subject-level cross-validation (CV). The average score and the standard deviation are both listed. To make our results comparable with previous papers, HCPA, HCPYA, and ADNI use a 5-fold CV as same as [9, 31], while others use 10-fold as same as [33]. Since HCPs are used for both pre-training and fine-tuning, the training data in the two stages is always from the corresponding CV fold's training set to prevent data leakage. Hyperparameters, e.g., learning rate and hidden channels, can be found in the Appendix.

State-of-the-art (SOTA) brain foundation models, BrainMass [34] and BrainJEPA [11], are selected as expert architectures along with the new classifier architecture proposed in this work. Note that the original BrainMass fine-tuning utilizes the support vector machine (SVM) that has a lower scale of learnable parameters than others. Therefore, an enhancement of BrainMass is evaluated in this work by replacing SVM with a 2-layer MLP.

Table 1: MoE improvement on phenotypic classification F1 score compared to the baseline, where 30k is pre-trained on resting-state data ($n$=29,951), and 68k is pre-trained on all data ($n$=68,251). PT stands for pre-training. Colored text indicates the performance increase/decrease from using 68k.

| Predictor
PT # | BrainMass | | | | | BrainJEPA | | |
|---|---|---|---|---|---|---|---|---|
| | SVM
30k | SVM
68k | MLP
30k | MLP
68k | BrainMoE
68k | ViT
30k | ViT
68k | BrainMoE
68k |
| ADNI | $75.32_{\pm7.06}$ | $75.32_{\pm7.06}$ | $76.86_{\pm7.26}$ | $80.70_{\pm7.85}$ | $81.23_{\pm11.00}$ | $74.16_{\pm8.55}$ | $74.16_{\pm8.55}$ | $77.11_{\pm6.64}$ |
| ↳AD | | 0.00 | | 3.84 ↑ | 4.37 ↑ | | 0.00 | 2.95 ↑ |
| ABIDE | $62.31_{\pm1.95}$ | $64.12_{\pm2.31}$ | $66.81_{\pm4.18}$ | $67.81_{\pm3.91}$ | $70.26_{\pm3.40}$ | $36.25_{\pm6.93}$ | $39.82_{\pm3.91}$ | $54.55_{\pm9.89}$ |
| ↳ASD | | 1.81 ↑ | | 1.00 ↑ | 3.45 ↑ | | 3.77 ↑ | 18.30 ↑ |
| PPMI | $54.87_{\pm15.76}$ | $56.52_{\pm14.86}$ | $58.90_{\pm14.29}$ | $59.77_{\pm14.22}$ | $62.97_{\pm13.94}$ | $38.69_{\pm13.91}$ | $38.69_{\pm13.91}$ | $60.49_{\pm11.59}$ |
| ↳PD (staged) | | 1.65 ↑ | | 0.87 ↑ | 4.07 ↑ | | 0.00 | 21.80 ↑ |
| Taowu | $58.33_{\pm34.78}$ | $65.67_{\pm20.55}$ | $70.29_{\pm17.97}$ | $68.00_{\pm21.46}$ | $88.57_{\pm12.51}$ | $36.08_{\pm26.38}$ | $36.94_{\pm20.98}$ | $79.86_{\pm14.46}$ |
| ↳PD (binary) | | 7.34 ↑ | | 2.29 ↓ | 18.28 ↑ | | 0.86 ↑ | 43.78 ↑ |
| SZ | $76.95_{\pm9.01}$ | $76.95_{\pm9.01}$ | $79.85_{\pm8.69}$ | $77.63_{\pm8.56}$ | $83.10_{\pm11.33}$ | $76.98_{\pm9.00}$ | $78.97_{\pm9.79}$ | $82.86_{\pm9.19}$ |
| ↳Schizophrenia | | 0.00 | | 2.22 ↓ | 3.25 ↑ | | 1.99 ↑ | 5.88 ↑ |
| HCPA | $85.16_{\pm0.41}$ | $89.73_{\pm0.58}$ | $87.91_{\pm0.48}$ | $90.63_{\pm0.74}$ | $96.67_{\pm0.77}$ | $59.54_{\pm15.47}$ | $53.12_{\pm14.19}$ | $81.74_{\pm0.51}$ |
| ↳3-task,rest | | 4.57 ↑ | | 2.72 ↑ | 8.76 ↑ | | 6.42 ↓ | 22.20 ↑ |
| HCPYA | $77.51_{\pm2.42}$ | $80.87_{\pm1.77}$ | $79.40_{\pm1.78}$ | $81.27_{\pm1.27}$ | $93.19_{\pm0.72}$ | $50.68_{\pm25.20}$ | $56.10_{\pm29.16}$ | $74.59_{\pm3.79}$ |
| ↳7-task | | 3.36 ↑ | | 1.87 ↑ | 13.79 ↑ | | 5.42 ↑ | 23.91 ↑ |

## 4.1 RQ1: MoE vs baselines

The average F1 scores of BrainMoE and BrainMass pretrained with all samples, and $n$=12 individual experts per cognitive state on phenotypic and sex classifications among 7 datasets are shown in Fig. 4. Previous studies have demonstrated good PT data scalability with resting-state fMRI data. However, according to Fig. 4, there are consistently existing task-specific experts (e.g., language for AD, working memory for PD) outperforming Resting experts, confirming that task-state fMRI contains valuable information for brain modeling. Conclusively, BrainMoE holds the best performance compared to all experts across 7 datasets. This supports that the utilization of cognitive embeddings from BrainMoE leads to more robustness of brain modeling than naively training a single task of fMRI.

In Table 1, we summarize the impact of BrainMoE on downstream phenotypic classification, reporting improvements in the F1 score relative to non–MoE baselines for disease and human behavior recognition. Across all tasks (ADNI, ABIDE, PPMI, Taowu, SZ, HCPA, HCPYA), intuitively expanding pre-training data from 30k to 68k yields modest gains, even negative gains, for both BrainMass with SVM and MLP, e.g. ABIDE BrainMass SVM: +1.81 F1 and SZ BrainMass MLP: -2.22 F1, and BrainJEPA with Vision Transformer (ViT), e.g., HCPA BrainJEPA: -6.42 F1. In contrast, introducing our BrainMoE with the proposed cognition adapter on top of the 68k pre-trained backbone amplifies these gains substantially: phenotypic F1 score uplifts range from +3.25 F1 (SZ Schizophrenia) to +43.78 F1 (Taowu PD), and it consistently brings a positive effect. Even BrainJEPA baselines that do not gain F1 improvement from more data on HCPA and HCPYA benefit from BrainMoE, albeit to a lesser extent (e.g., HCPYA: +3.04 F1). Notably, the largest relative benefit appears on smaller cohorts, e.g., Taowu ($n$=40), where BrainMoE achieves +18.28 and +43.76 F1 over 68k BrainMass and BrainJEPA baselines.

Table 2 reports analogous results for sex classification. Worse than phenotypic classification, increasing pre-training size delivers small, commonly no improvements for BrainMass and BrainJEPA using SVM, MLP, and ViT, where 11 out of 18 experiments have dropped F1 scores in red text. In contrast, BrainMoE recovers and exceeds prior performance. It consistently demonstrates F1 score gains, except for BrainJEPA on ABIDE. It is worth noting that the most dramatic uplift appears on the smallest dataset (Taowu, $n$=40), where BrainMoE increases F1 by +43.76 on BrainJEPA.

Overall, the above results demonstrate that (1) a large scale pre-training without stratifying cognitive states improves downstream performance modestly (sometimes negatively), and (2) the proposed BrainMoE framework produces substantial gains, especially on downstream applications with a limited sample size.

Table 2: MoE improvement on sex classification F1 score compared to the baseline, where the sample size of the downstream dataset is indicated. PT stands for pre-training.

| Predictor | BrainMass | | | | | BrainJEPA | | |
|---|---|---|---|---|---|---|---|---|
| | SVM | SVM | MLP | MLP | BrainMoE | ViT | ViT | BrainMoE |
| PT # | 30k | 68k | 30k | 68k | 68k | 30k | 68k | 68k |
| ADNI | $48.60_{\pm6.55}$ | $54.30_{\pm12.48}$ | $64.82_{\pm4.30}$ | $59.30_{\pm13.05}$ | $69.22_{\pm5.26}$ | $37.70_{\pm8.73}$ | $36.42_{\pm8.22}$ | $62.98_{\pm7.76}$ |
| ↳n=138 | | 5.70 ↑ | | 5.52 ↓ | 4.40 ↑ | | 1.28 ↓ | 25.28 ↑ |
| ABIDE | $73.84_{\pm3.49}$ | $73.84_{\pm3.49}$ | $75.12_{\pm5.27}$ | $75.12_{\pm6.32}$ | $85.21_{\pm3.77}$ | $78.08_{\pm5.84}$ | $78.08_{\pm5.84}$ | $78.08_{\pm6.15}$ |
| ↳n=1025 | | 0.00 | | 0.00 | 10.09 ↑ | | 0.00 | 0.00 |
| PPMI | $52.03_{\pm14.16}$ | $56.58_{\pm12.28}$ | $63.32_{\pm14.96}$ | $64.73_{\pm14.12}$ | $79.81_{\pm8.46}$ | $46.23_{\pm9.00}$ | $46.23_{\pm9.00}$ | $67.57_{\pm7.24}$ |
| ↳n=209 | | 4.55 ↑ | | 1.41 ↑ | 16.49 ↑ | | 0.00 | 21.34 ↑ |
| Taowu | $46.24_{\pm23.96}$ | $46.24_{\pm23.96}$ | $62.86_{\pm28.20}$ | $55.38_{\pm20.93}$ | $74.00_{\pm27.79}$ | $46.24_{\pm23.97}$ | $51.67_{\pm21.12}$ | $90.00_{\pm12.96}$ |
| ↳n=40 | | 0.00 | | 7.48 ↓ | 11.14 ↑ | | 5.43 ↑ | 43.76 ↑ |
| HCPA | $66.20_{\pm1.58}$ | $68.25_{\pm1.70}$ | $66.93_{\pm0.63}$ | $68.58_{\pm0.99}$ | $76.76_{\pm0.93}$ | $40.32_{\pm4.07}$ | $40.32_{\pm4.07}$ | $44.24_{\pm7.01}$ |
| ↳n=4863 | | 2.05 ↑ | | 1.65 ↑ | 9.83 ↑ | | 0.00 | 3.92 ↑ |
| HCPYA | $63.33_{\pm3.01}$ | $65.47_{\pm3.51}$ | $64.57_{\pm2.53}$ | $66.98_{\pm3.30}$ | $74.36_{\pm4.43}$ | $40.20_{\pm4.22}$ | $40.20_{\pm4.22}$ | $43.24_{\pm8.19}$ |
| ↳n=3293 | | 2.14 ↑ | | 2.41 ↑ | 9.79 ↑ | | 0.00 | 3.04 ↑ |

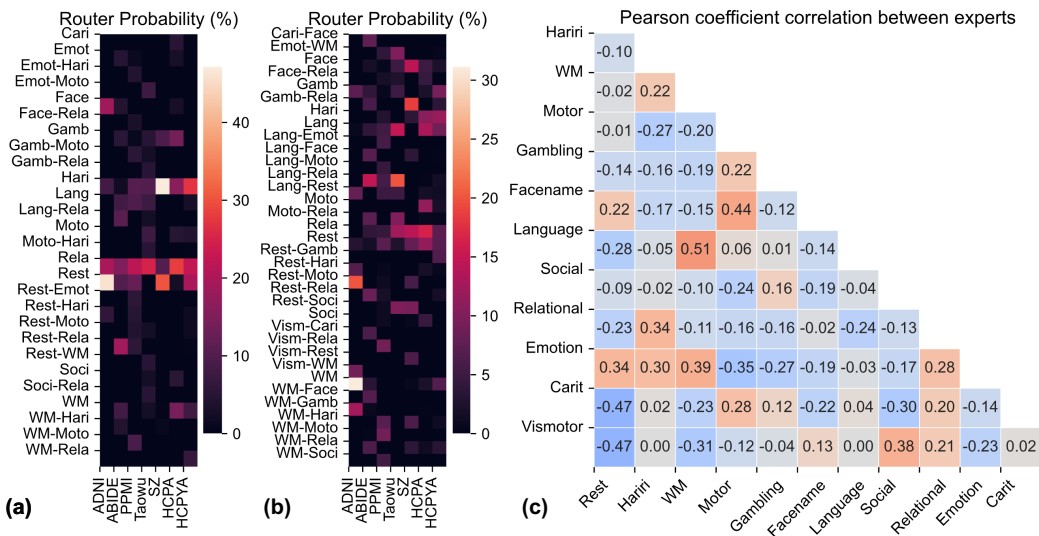

Figure 5: The distribution of dominating expert combinations of **(a)** late fusion MoE and **(b)** BrainMoE shows the router preference, where the first 4 letters of cognitive state are used as abbreviation. **(c)** The correlation between cognition embeddings of experts shows expert diversity.

## 4.2 RQ2: Pre-training objectives

As described in Sec 3, the BrainMoE framework has no requirement for data type and architecture. This property results in various pre-training objectives for the brain expert: FC reconstruction (FC recon.), BOLD reconstruction (BOLD recon.), and cognitive state classification (Cog. Classif.). To demonstrate which objectives are the most robust, we evaluated three objectives and an all-in-one BrainMoE on 7 downstream datasets in Table 3 and 4.

Briefly, FC reconstruction and all-in-one BrainMoE show the best robustness. They always rank in the first two places for phenotypic (Table 3) and sex classification (Table 4) across all downstream datasets, except for the smallest dataset Taowu (n=40). Unlike BrainMoE has the cognition adapter to implicitly utilize expert embeddings, the late fusion explicitly combines expert predictions, therefore cannot handle the unbalance issue. We can observe in Table 3 that FC reconstruction shows the most first rank, 4 out of 7 datasets, and all-in-one has the most first place, 4 out of 7, in Table 4. Although top-$k$ selected experts in all-in-one BrainMoE contain experts in FC reconstruction, the self-attention in the cognition adapter mixes information between all types of pre-training objectives, yielding dropped and boosted performance for phenotypic and sex classification, respectively.

Table 3: MoE performance on phenotypic classification F1 score using three types of expert pre-trained with three objectives, along with an all-in-one MoE mixing all types of experts, where LF is Late Fusion, Ex. # denotes expert number. **Bold** is the first rank and underline is the second.

| | Ex. # | ADNI | ABIDE | PPMI | Taowu | SZ | HCPA | HCPYA |
|---|---|---|---|---|---|---|---|---|
| **Baseline** | | | | | | | | |
| BrainMass | 1 | $80.70_{+7.85}$ | $67.81_{+3.91}$ | $59.77_{+14.22}$ | $68.00_{+21.46}$ | $77.63_{+8.56}$ | $90.63_{+0.74}$ | $81.27_{+1.27}$ |
| BrainJEPA | 1 | $74.16_{+8.55}$ | $39.82_{+3.91}$ | $38.69_{+13.91}$ | $36.94_{+20.98}$ | $78.97_{+9.79}$ | $53.12_{+14.19}$ | $56.10_{+29.16}$ |
| LF-MoE | 12 | $73.33_{+10.87}$ | $\underline{69.89}_{+3.06}$ | $\underline{61.11}_{+15.29}$ | $\mathbf{91.24}_{+11.72}$ | $76.95_{+9.50}$ | $94.96_{+2.64}$ | $88.58_{+3.39}$ |
| **BrainMoE** | | | | | | | | |
| FC recon. | 12 | $\mathbf{81.23}_{+11.00}$ | $\mathbf{70.26}_{+3.40}$ | $\mathbf{62.97}_{+13.94}$ | $88.57_{+12.51}$ | $83.10_{+11.33}$ | $\mathbf{96.67}_{+0.77}$ | $93.19_{+0.72}$ |
| BOLD recon. | 12 | $77.11_{+6.64}$ | $54.55_{+9.89}$ | $60.49_{+11.59}$ | $79.86_{+14.46}$ | $82.86_{+9.19}$ | $81.74_{+0.51}$ | $74.59_{+3.79}$ |
| Cog. classif. | 12 | $79.70_{+10.28}$ | $68.65_{+3.81}$ | $59.23_{+14.65}$ | $90.48_{+14.64}$ | $\mathbf{83.36}_{+10.08}$ | $96.28_{+0.70}$ | $95.81_{+0.48}$ |
| All-in-one | 36 | $\underline{79.73}_{+10.60}$ | $69.13_{+4.08}$ | $60.76_{+14.85}$ | $85.93_{+18.32}$ | $\underline{83.91}_{+8.07}$ | $\underline{96.66}_{+0.94}$ | $\mathbf{96.81}_{+0.41}$ |

Table 4: MoE performance on sex classification F1 score using three types of expert pre-trained with three objectives, along with an all-in-one MoE mixing all types of experts. **Bold** is the first rank and underline is the second.

| | Ex. # | ADNI | ABIDE | PPMI | Taowu | HCPA | HCPYA |
|---|---|---|---|---|---|---|---|
| **Baseline** | | | | | | | |
| BrainMass | 1 | $59.30_{+13.05}$ | $75.12_{+6.32}$ | $64.73_{+14.12}$ | $55.38_{+20.93}$ | $68.58_{+0.99}$ | $66.98_{+3.30}$ |
| BrainJEPA | 1 | $36.42_{+8.22}$ | $78.08_{+5.84}$ | $46.23_{+9.00}$ | $51.67_{+21.12}$ | $40.32_{+4.07}$ | $40.20_{+4.22}$ |
| **BrainMoE** | | | | | | | |
| FC recon. | 12 | $69.22_{+5.26}$ | $\mathbf{85.21}_{+3.77}$ | $\underline{79.81}_{+8.46}$ | $74.00_{+27.79}$ | $76.76_{+0.93}$ | $74.36_{+4.43}$ |
| BOLD recon. | 12 | $\underline{62.98}_{+7.76}$ | $78.08_{+6.15}$ | $67.57_{+7.24}$ | $\mathbf{90.00}_{+12.96}$ | $44.24_{+7.01}$ | $43.24_{+8.19}$ |
| Cog. classif. | 12 | $65.75_{+7.61}$ | $82.82_{+5.11}$ | $79.07_{+7.28}$ | $72.22_{+25.17}$ | $75.65_{+1.26}$ | $\underline{75.18}_{+1.15}$ |
| All-in-one | 36 | $\mathbf{70.72}_{+7.58}$ | $\underline{82.85}_{+5.91}$ | $\mathbf{82.80}_{+5.65}$ | $\underline{75.29}_{+30.56}$ | $\mathbf{78.34}_{+2.18}$ | $\mathbf{77.67}_{+1.54}$ |

## 4.3 Rounter and expert analysis

The preference of routers in late fusion MoE and BrainMoE is shown in Fig. 5 (a) and (b), respectively, where the percentage indicates how many samples have a combination of experts with dominating router logits ($\geqslant \frac{1}{N}$). Clearly, BrainMoE has diverse and similar dominating combinations for heterogeneous and homogeneous applications, respectively. The combinations are mainly dual, and datasets with the same task share similar pattern (i.e., cognitive state for HCPA and HCPYA, and PD for PPMI and Taowu). In contrast, late fusion consistently has single dominating expert due to data scale diversity, e.g., Rest ($n$=29,971), which implies that the routers trained by BrainMoE learned more neuroscientific knowledge than the late fusion. Furthermore, the investigation on expert embeddings is shown in Fig. 5 (c). The absolute value of correlation is mostly less than 0.5, indicating experts are not in conflict or redundant to each other.

## 4.4 More applications

The age regression across 6 datasets has been evaluated for the best baseline, 68k version of BrainMass, and BrainMoE, as listed in Table 5. We can observe the performance of BrainMoE is the best, where the improvement is especially significant for ABIDE (n=1,025, 6-58 yrs) with MSE 36.77 $\longrightarrow$ 4.86. This empirical evidence further supports the robustness of BrainMoE.

Table 5: Age regression performance compared to the baseline, where the sample size of the downstream dataset is indicated, and unit is year.

| MSE | ADNI | ABIDE | PPMI | Taowu | HCPA | HCPYA |
|---|---|---|---|---|---|---|
| BrainMass | $36.28_{+19.83}$ | $36.77_{+17.2}$ | $33.09_{+21.87}$ | $38.10_{+28.33}$ | $22.66_{+7.51}$ | $5.46_{+2.69}$ |
| BrainMoE | $\mathbf{36.27}_{+10.23}$ | $\mathbf{4.86}_{+2.67}$ | $\mathbf{29.89}_{+8.39}$ | $\mathbf{30.72}_{+12.25}$ | $10.56_{+2.86}$ | $\mathbf{3.45}_{+1.08}$ |

Multimodal applications of cognitive state, sex classifications, and age regression in an fMRI-EEG dataset [27] are listed in Table 6. There are 388 fMRI-EEG pairs from 22 healthy subjects under 8 cognitive states (age$\in$[23,51], F:M=1:1). BrainMoE is evaluated here with $n$=1 pre-

Table 6: BrainMoE applies on a multimodal dataset, NATVIEW [27].

| NATVIEW | 8-task (F1) | Sex (F1) | Age (MSE) |
|---|---|---|---|
| BrainMass (fMRI) | $67.66_{+5.74}$ | $63.67_{+5.16}$ | $8.05_{+5.58}$ |
| CBraMod (EEG) | $68.71_{+1.46}$ | $65.39_{+2.33}$ | $8.26_{+5.97}$ |
| BrainMoE (13 Ex.) | $\mathbf{68.73}_{+3.72}$ | $\mathbf{65.47}_{+5.38}$ | $\mathbf{7.99}_{+5.53}$ |

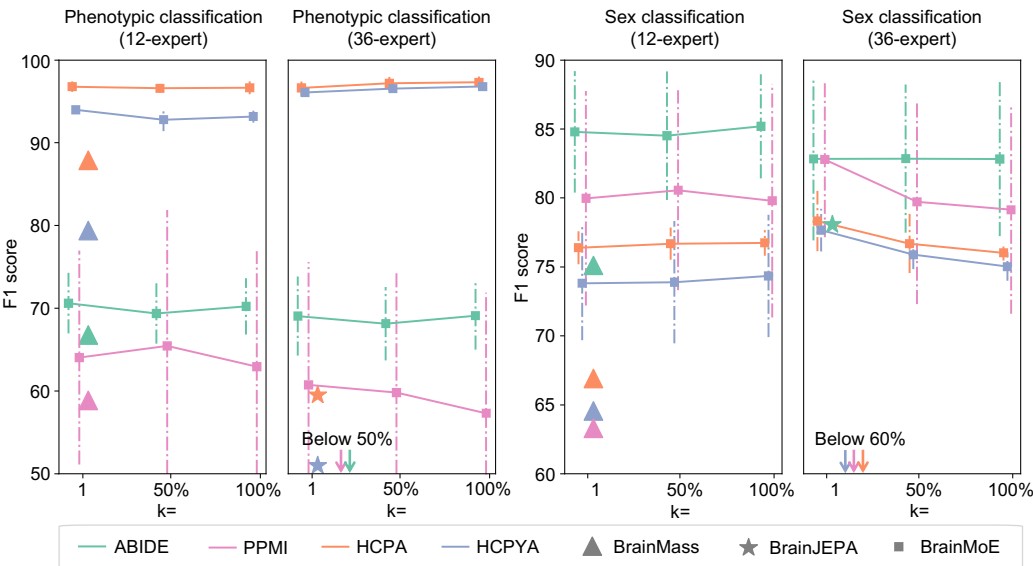

Figure 6: Impact of altering $k$ on downstream classification performance. Mean F1 scores with standard deviation are reported for four downstream datasets under three expert-selection regimes (top-1, top 50%, and all experts). Colors represent different datasets.

trained CBraMod [30] expert plus $n$=12 cognition classifier experts (Shaefer400 version). Results from a 5-fold CV show the best performance by multimodal BrainMoE.

## 4.5 Ablations

**Top-$k$** We alter $k$ in BrainMoE to strictly limit how many top experts can be selected for downstream applications. We evaluated $k = 1$, $k = 50\%$, and $k = 100\%$ with 12 and 36 experts on four datasets with relatively larger sample sizes in Fig. 6, where 12-expert BrainMoE uses BrainMass as the expert architecture. From the left two panels, it is obvious that increasing $k$ has a slight difference for all phenotypic tasks, except for HCPA. In the right two panels, we can observe that sex classification benefits less from a larger $k$. Overall, expert scaling in BrainMoE yields clear benefits for complex, data-scarce phenotypic tasks, with most gains achieved by employing top half of the experts. Beyond this, adding experts delivers diminishing returns. In contrast, for the simpler binary sex classification task, especially with a large expert pool, additional experts do not meaningfully improve performance and may introduce redundancy or overfitting. Thus, tailoring the number of active experts to task complexity and dataset size is key for efficient MoE deployment.

## 5 Conclusion

In conclusion, we propose a new framework of the brain foundation model, BrainMoE, to pre-train with overlooked tasking-state fMRI for robust downstream applications. We observe that existing brain foundation models learn from fMRI derived from a narrow range of cognitive states, while there are 11 available cognitive states as subjects performing explicit tasks in large scale datasets. Furthermore, we showcase that (i) the straightforward utilization of data with rich human behavioral variables by pre-training with all data and (ii) the late fusion MoE both improve performance marginally. Aiming at these challenges, BrainMoE pre-trains each expert on a portion of the datasets with the same cognitive state among 12 different states for a robust brain foundation model. We design a scalable cognition adapter to mix brain experts for downstream fine-tuning so that BrainMoE can handle orthogonal cognition embeddings and be robust on the boutique downstream datasets. With sufficient 68,251 pre-training fMRI scans among UKB and HCP with 12 different cognitive states, BrainMoE has shown impressive performance boosting on a variety of applications, including sex, age prediction, human behavior recognition, multimodal applications, and early diagnosis of various brain diseases. The promising results demonstrated on eight datasets from three different pipelines indicate great potential to facilitate current neuroimaging applications in clinical routines.

## Acknowledgement

This work was supported by the National Institutes of Health (AG091653, AG068399, AG084375) and the Foundation of Hope. Tianlong Chen was partially funded by the National Institutes of Health (NIH) under award 1R01EB037101-01. The views and conclusions contained in this document are those of the authors and should not be interpreted as representing the official policies, either expressed or implied, of the NIH.

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

## A    Accessibility

Public data is accessible via internet (UKB[2], HCPA[3], HCPYA[4], ADNI[5]. PPMI, ABIDE, and Taowu can be found here[6]). The licenses to obtain those data can also be accessed on the websites. The codes and data split settings can be acquired via this code repository[7].

## B    Data preprocessing

The neuroimage processing used for ADNI, UKB, HCPYA, and HCPA consists of the following major steps: (1) We segment the T1-weighted image into white matter, gray matter, and cerebral spinal fluid using FSL software [16]. (2) On top of the tissue segmentation in Fig. 7, we parcellate the cortical surface of fMRI into cortical regions according to the atlas as a regional signal of time-series in Fig. 7, where FC, in the end, is the Pearson correlation coefficient between regional time-series.

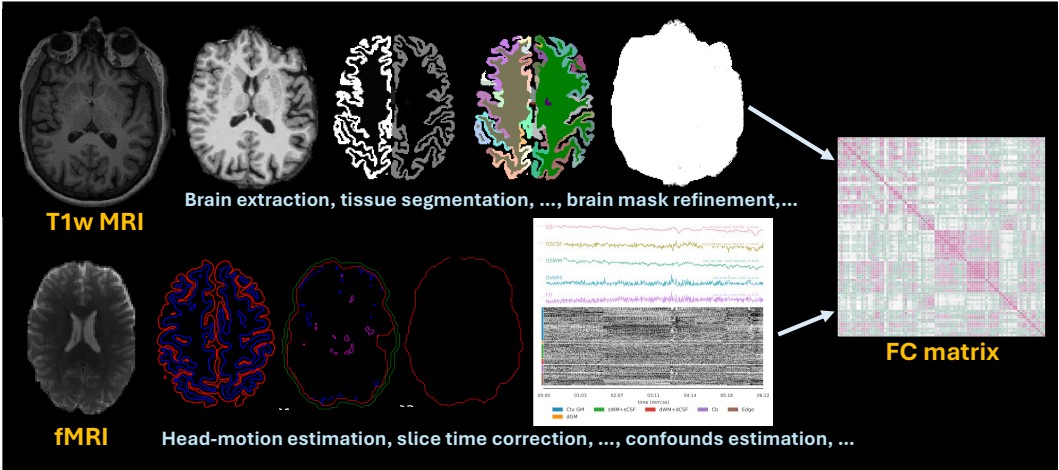

Figure 7: General workflows for processing T1-weighted image (T1w MRI) and functional MRI (fMRI). The output is shown at the right, including the brain network of FC.

## C    Computing environments and hyperparameters

The experiments are done on a Linux system with one NVIDIA RTX 6000 Ada. Batch size and learning rate are set as 128 and 1e-4, respectively. The maximum epoch is set as 200 and $C_{hid} = 2048$. Training will be early stopped if accuracy keeps dropping in 50 epochs.

## D    Comparison between previous works

We list the comparison of experimental datasets between previous works in Table 7.

## E    Computational complexity

The limitation of BrainMoE is more computational complexity, as listed in Table 8. BrainMoE with 12 experts spends $4\times$ more time than baselines. All-in-one with 36 experts nearly doubles the time cost.

---

[2]https://www.ukbiobank.ac.uk/

[3]https://www.humanconnectome.org/

[4]https://www.humanconnectome.org/study/hcp-young-adult/overview

[5]https://adni.loni.usc.edu/

[6]https://auckland.figshare.com/articles/dataset/NeurIPS_2022_Datasets/21397377

[7]https://github.com/Chrisa142857/brain_moe

Table 7: The comparison of experimental datasets between previous works.

|  | BrainLM (2024) [23] | BrainMass (2024) [34] | BrainJEPA (2024) [11] | BrainMoE (Ours) |
|---|---|---|---|---|
| Brain atlas | AAL424 | C200 | Schaefer400 | AAL116 |
| Cognitive state | resting, task-hariri | resting | resting | resting, 11 types of tasking |
| Pre-train dataset | UKB, HCP | UKB, HCP, OpenNeuron | UKB, HCP | UKB, HCP |
| Pre-train data # | 61,038 | 64,584 | 40,162 | 68,251 |
| Fine-tune dataset | UKB, HCP | ASD, ADHD, AD, PD, MDD | UKB, HCP, ADNI | HCP, ASD, AD, PD, SZ |
| Parameter amount | 650M | 34M | 307M | 709M |

Table 8: Computational time cost of BrainMoE inference with two existing architectures and the all-in-one BrainMoE on the ABIDE dataset.

| Test time (ms/sample) | BrainMass | BrainJEPA |
|---|---|---|
| Single model | 37.08 | 28.13 |
| BrainMoE | 157.60 | 133.26 |
| All-in-one | 287.21 | |

## F   Visual decoding potential

Visual decoding task for a new dataset NSD [2] has also been evaluated for MindEye2 [26] as the baseline and BrainMoE. We pre-trained two MindEye2s as the specific experts for long-term (novel trials) and short-term memory (easy/hard trials), respectively. Since visual decoding is a generative task, output contains much higher dimensions ($256 \times 1664$ vs. class number 2 to 7) than downstream tasks focused in the main text. Therefore, we skipped our cognition adapter by weighted summing the diffusion prior of two experts with the BrainMoE routing probabilities. Both baseline and BrainMoE are pretrained with subjects 2-7 and finetuned with subject 1 on the entire 40 sessions. The final train and test losses, cosine similarity, and Mean Squared Error (MSE) during finetuning are listed in Table 9. Given the evidence that the performance of BrainMoE is better than the single expert MindEye2, there is potential for BrainMoE to expand to visual decoding.

Table 9: Visual decoding performance.

|  | MindEye2 | BrainMoE |
|---|---|---|
| Train loss | 9.639 | **7.994** |
| Test loss | 11.142 | **9.405** |
| Cos. Sim. | 0.778 | **0.840** |
| MSE | 0.301 | **0.261** |

## G   Scalability analysis

MLP as a universal predictive head is used in related works for the MoE adapter. Fig. 8 is the comparison between MLP and the proposed cognition adapter with different amounts of learnable parameters.

## H   Visualization

The attention weights conducted by BrainMoE with different $k$ is visualized in Fig. 9. We can observe: (1) Advanced by the cognition adaptor, BrainMoE agrees with current neuroscience knowledge since it mainly attends to DAN and DMN for ASD [12, 21], SMN and FPN for PD [7]. (2) Differences are slight across enlarged $k$, indicating that the router produces consistent expert weights.

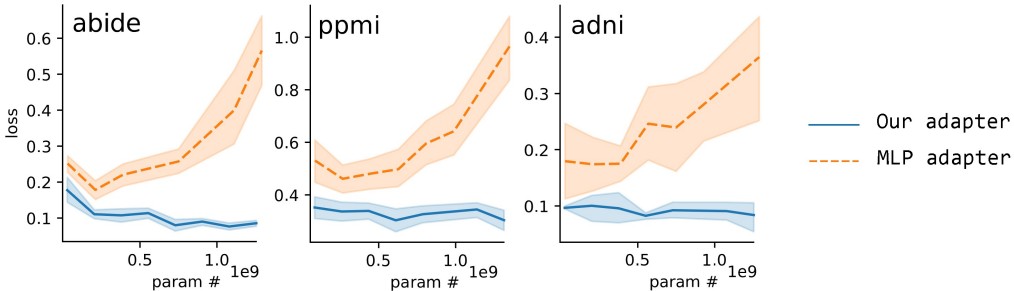

Figure 8: The scalability of the MLP adapter and our adapter on disease prediction, where the y-axis is the fine-tuning loss.

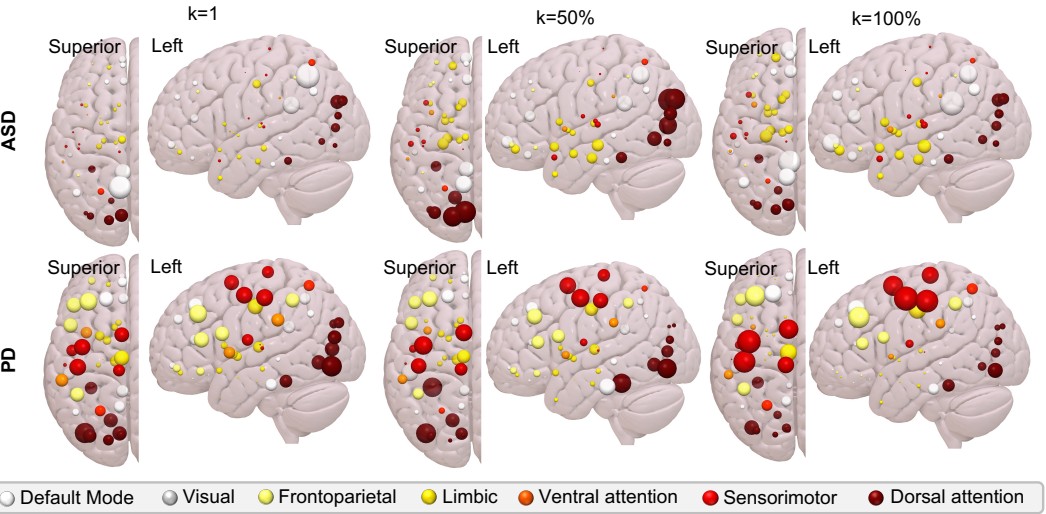

Figure 9: Visualization of attention weights by FC reconstruction BrainMoE.

