# OpenReview forum: "BrainMoE: Cognition Joint Embedding via Mixture-of-Expert Towards Robust Brain Foundation Model"
_NeurIPS.cc/2025/Conference — NeurIPS 2025 poster_

### Official Review · Reviewer_Yxzq · 2025-06-29

**Clarity:** 4
**Significance:** 3
**Originality:** 3
**Rating:** 5
**Confidence:** 4

**Summary:**

The idea is to use separate experts for separate cognitive states/tasks. In addition to MSE the authors also use cognitive state classification. Traditional models use the same weights across wildly different fMRI tasks. The model demonstrates substantial performance gains over baselines across 7 datasets and 3 preprocessing pipelines.

**Questions:**

How do we know the performance boost is not due to increase in parameters?
How do you think this is better compared to task embeddings?
Why was a Transformer decoder chosen over other scalable alternatives like MLPs or lightweight convolutional heads?
Did the authors observe any conflicts or redundancy between experts trained on similar cognitive states?
Could BrainMoE generalize to modalities beyond fMRI (e.g., EEG) if modality-specific experts are trained?

**Ethical Concerns:**

["NO or VERY MINOR ethics concerns only"]

**Final Justification:**

I maintain my original acceptance recommendation, as it is already high, and suggesting a strong accept would need groundbreaking results with much higher impact.

**Limitations:**

yes

**Paper Formatting Concerns:**

Many grammar errors throughout the paper.

The chosen color in tables is jarring (green)

**Quality:**

3

**Strengths And Weaknesses:**

### Strength
- This is a novel application of mixture-of-experts to fMRI data.
- Shows considerable performance improvement in multiple datasets/tasks
- Good evaluation of different pretraining objectives
- Compatible with existing foundational models and input types
- Good reproducibility


###  Weaknesses
- Evaluation should be done primarily cross-subject
- Higher computational cost

---

> ### Author Rebuttal · Authors · 2025-07-31
>
> ## **W1**
>
> Thank you for the comment. Our evaluations are exactly cross-subject. Each fMRI in the five clinical datasets corresponds to one unique subject, and the evaluations are subject-wise (L.191-L.206). Since two non-clinical datasets are for cognitive state recognition, there are multiple cognitive states per subject, but the train/test split in our multi-fold cross-validation is still subject-level (L.207).
>
> ## **W2**
>
> The running time of BrainMoE has fulfilled the real-time requirement for fMRI applications, which takes a much longer time (10-15mins) than BrainMoE (133-287ms) for data acquisition using a common 3T scanner [1].
> Additionally, BrainMoE has been implemented with the pre-computation of cognitive embeddings from complex experts to amortize the high computational costs from multiple brain experts. For instance, BrainJEPA experts have high FLOPs (5.32T) and compute cognitive embeddings prior to BrainMoE finetuning, while BrainMass experts with much lower FLOPs (26.96M) have the cognitive embeddings computed on-the-fly. This trick enhanced efficiency as listed in Table 5, where BrainJEPA with the trillion-level FLOPs costs less time than BrainMass with the million level FLOPs. We will include the discussion of this possible amortized efficiency pruning in the “Limitation” section of the final version.
>
> [1] Kumar, V. A., et al. "Recommended resting-state fMRI acquisition and preprocessing steps for preoperative mapping of language and motor and visual areas in adult and pediatric patients with brain tumors and epilepsy." American Journal of Neuroradiology 45.2 (2024): 139-148.
>
> ## **Q1**
>
> Parameters are increased $N$ times compared to the previous single expert model, where $N$ is the number of experts. However, increasing the parameter amount by fusing multiple experts naively, e.g., a late fusion MoE [2], sometimes cannot boost the robustness of brain modeling (80.70 -> 73.33 for AD prediction). The table below shows the seven downstream applications using baseline (BrainMass), the late fusion MoE, and our BrainMoE with the parameter amount of each model. BrainMoE, in contrast, consistently has the best performance.
>
> Note that late fusion is a strategy for handling multimodal data where the final output is the weighted sum of predictions from each independent expert selected by the router. Unlike BrainMoE has the cognition adapter to implicitly utilize multiple expert embeddings via Transformer, the late fusion directly combining expert task embeddings suffers from a data unbalance issue for different domain experts (Table A for Reviewer ZhRL). Therefore, the performance boost of BrainMoE is not only advanced by more parameters but also by our new ML component, the cognition adapter.
>
> ||ADNI (AD)|ABIDE (ASD)|PPMI (PD)|Taowu (PD)|SZ (SZ)|HCPA (3-task)|HCPYA (7-task)|
> |-|-|-|-|-|-|-|-|
> |Baseline (34M)|80.70$_{\\pm7.85}$|67.81$_{\\pm3.91}$|59.77$_{\\pm14.22}$|68.00$_{\\pm21.46}$|77.63$_{\\pm8.56}$|90.63$_{\\pm0.74}$|81.27$_{\\pm1.27}$|
> |Late fusion MoE (408M)|73.33$_{\\pm10.87}$|69.89$_{\\pm306}$|61.11$_{\\pm15.29}$|**91.24$_{\\pm11.72}$**|76.95$_{\\pm9.50}$|94.96$_{\\pm2.64}$|88.58$_{\\pm3.39}$|
> |BrainMoE (709M)|**81.23$_{\\pm11.00}$**|**70.26$_{\\pm3.40}$**|**62.97$_{\\pm13.94}$**|88.57$_{\\pm12.51}$|**83.10$_{\\pm11.33}$**|**96.67$_{\\pm0.77}$**|**93.19$_{\\pm0.72}$**|
>
> [2] Han, Xing, et al. "Fusemoe: Mixture-of-experts transformers for fleximodal fusion." Advances in Neural Information Processing Systems 37 (2024): 67850-67900.
>
> ## **Q2**
>
> BrainMoE approach uses cognitive embeddings so that the cognition adapter can learn from the latent space, which is the information bottleneck between data space and task space (Fig. 1). The latent space is a lower-dimensional representation of the input data, is designed to capture the most essential and relevant information from the original data while discarding less important details [3]. The task embeddings, on the other hand, capture the characteristics of a specific downstream application instead of various cognition-related features in the latent space by domain experts, while either clinical or non-clinical brain modeling is highly relevant to cognitive function or dysfunction. The less robust performance of late fusion MoE further provides empirical evidence to support that cognitive embeddings are better than task embeddings.
>
> [3] Wang, Shiye, et al. "Self-supervised information bottleneck for deep multi-view subspace clustering." IEEE Transactions on Image Processing 32 (2023): 1555-1567.
>
> ## **Q3**
>
> The scalability of MLP has been analyzed in Appendix Figure 8. Curves of the train loss at the 200th epoch by MLP are increasing with more parameters, using more layers and hidden channels in 3 datasets, demonstrating the bad scalability in brain modeling. In contrast, our Transformer-based adapter maintains a decreasing train loss at the 200th epoch when there are more parameters. Note that lightweight convolutional heads are specific for image data, while FC is structured data, and BOLD is a timeseries data, so that convolutional heads cannot be adapted to our cases.
>
> ## **Q4**
>
> No, there are no observable identical or conflicting cognitive embeddings from different brain experts in the observation of the correlation between cognitive embeddings. The table below is the mean value of Pearson’s correlation between BrainMass experts specified for 12 different cognitive states, where the maximum correlation is 0.51 [Language, Working memory] and the minimum is -0.47 [Vismotor, Rest]. The absolute value of correlation is mostly less than 0.5, indicating experts are not in conflict or redundant to each other.
>
> |Expert pair|Mean corr|Expert pair|Mean corr|Expert pair|Mean corr|Expert pair|Mean corr|
> |-|-|-|-|-|-|-|-|
> |vismotor-carit|0.02|carit-facename|\-0.22|rest-gambling|\-0.14|wm-social|\-0.1|
> |vismotor-language|0|carit-motor|0.28|rest-facename|0.22|wm-hariri|0.22|
> |vismotor-rest|**\-0.47**|carit-social|\-0.3|rest-motor|\-0.01|wm-relational|\-0.11|
> |vismotor-emotion|\-0.23|carit-hariri|0.02|rest-social|\-0.09|gambling-facename|\-0.12|
> |vismotor-wm|\-0.31|carit-relational|0.2|rest-hariri|\-0.1|gambling-motor|0.22|
> |vismotor-gambling|\-0.04|language-rest|\-0.28|rest-relational|\-0.23|gambling-social|0.16|
> |vismotor-facename|0.13|language-emotion|\-0.03|emotion-wm|0.39|gambling-hariri|\-0.16|
> |vismotor-motor|\-0.12|language-wm|**0.51**|emotion-gambling|\-0.27|gambling-relational|\-0.16|
> |vismotor-social|0.38|language-gambling|0.01|emotion-facename|\-0.19|facename-motor|0.44|
> |vismotor-hariri|0|language-facename|\-0.14|emotion-motor|\-0.35|facename-social|\-0.19|
> |vismotor-relational|0.21|language-motor|0.06|emotion-social|\-0.17|facename-hariri|\-0.17|
> |carit-language|0.04|language-social|\-0.04|emotion-hariri|0.3|facename-relational|\-0.02|
> |carit-rest|\-0.47|language-hariri|\-0.05|emotion-relational|0.28|motor-social|\-0.24|
> |carit-emotion|\-0.14|language-relational|\-0.24|wm-gambling|\-0.19|motor-hariri|\-0.27|
> |carit-wm|\-0.23|rest-emotion|0.34|wm-facename|\-0.15|motor-relational|\-0.16|
> |carit-gambling|0.12|rest-wm|\-0.02|wm-motor|\-0.2|social-hariri|\-0.02|
> |social-relational|\-0.13|||||||
> |hariri-relational|0.34|||||||
>
> ## **Q5**
>
> Yes, it could. We selected a multimodal (fMRI+EEG) visual stimuli dataset [4] for the evaluation. There are 388 fMRI-EEG pairs from 22 healthy subjects under 8 cognitive states (age$\in$[23,51], F:M=1:1). The EEG expert is implemented with the pre-trained CBraMod [5]. Since the preprocessed fMRI in this dataset was using the Shaefer atlas and only Cog. Classif. experts in our work have been pretrained with Shaefer400, BrainMoE was evaluated on this FMRIEEG dataset with one CBraMod expert and 12 Cog. Classif. experts. This experiment is a subject-level 5-fold cross-validation. Results of cognitive state classification, sex classification, and age regression are shown in the tables below. Better performance by multimodal BrainMoE further indicates the value of our approach.
>
> The table below shows the cognitive state classification performance of BrainMoE, where 'Prec' is precision, and 'Rec' is recall.
>
> ---
> *cognitive state*
>
> |Expert|Accuracy|F1|Prec|Rec|AUC|
> |-|-|-|-|-|-|
> |fMRI|67.72$_{\pm5.50}$|67.66$_{\pm5.74}$|73.14$_{\pm6.44}$|67.72$_{\pm5.50}$|**90.45$_{\pm6.54}$**|
> |EEG|69.20$_{\pm2.19}$|68.71$_{\pm1.46}$|71.35$_{\pm1.20}$|**69.24$_{\pm2.19}$**|84.22$_{\pm1.16}$|
> |fMRI + EEG|**69.21$_{\pm4.06}$**|**68.73$_{\pm3.72}$**|**73.41$_{\pm1.99}$**|69.21$_{\pm4.06}$|87.09$_{\pm5.51}$|
>
> The table below shows the sex prediction performance of BrainMoE.
>
> ---
> *sex*
>
> |Expert|Accuracy|F1|Prec|Rec|AUC|
> |-|-|-|-|-|-|
> |fMRI|63.43$_{\pm4.60}$|63.67$_{\pm5.16}$|65.61$_{\pm5.84}$|63.43$_{\pm4.60}$|63.40$_{\pm7.39}$|
> |EEG|65.53$_{\pm3.19}$|65.39$_{\pm2.33}$|67.91$_{\pm3.32}$|65.53$_{\pm3.19}$|**65.68$_{\pm7.71}$**|
> |fMRI + EEG|**66.47$_{\pm3.54}$**|**65.47$_{\pm5.38}$**|**70.29$_{\pm4.70}$**|**66.47$_{\pm3.54}$**|65.00$_{\pm6.10}$|
>
> The table below shows the age regression performance of BrainMoE, where 'Abs. Corr.' is the absolute Pearson's correlation between ground truths and predictions, and 'MSE' is the mean squared error.
>
> ---
> *age*
>
> |Expert|MSE|Abs. Corr.|
> |-|-|-|
> |fMRI|8.05$_{\pm5.58}$|0.512$_{\pm0.204}$|
> |EEG|8.26$_{\pm5.97}$|0.475$_{\pm0.179}$|
> |fMRI + EEG|**7.99$_{\pm5.53}$**|**0.523$_{\pm0.152}$**|
>
> [4] Telesford, Qawi K., et al. "An open-access dataset of naturalistic viewing using simultaneous EEG-fMRI." Scientific Data 10.1 (2023): 554.
>
> [5] J. Wang et al., ‘CBraMod: A Criss-Cross Brain Foundation Model for EEG Decoding’, in The Thirteenth International Conference on Learning Representations, 2025.
>
> ## **Paper format 1**
>
> We appreciate your suggestion, and we are committed to thoroughly revising the manuscript. Specifically, we will highlight the innovation of BrainMoE and provide more comprehensive supporting results as shown in this rebuttal.
>
>
> ## **Paper format 2**
>
> We will revise the font color to dark green.

---

> ### Comment · Reviewer_Yxzq · 2025-08-01
>
> Thank you for the detailed responses. I agree with most points and as in my original review I think this paper should be accepted. Thanks for clarifying the cross-subject evaluation also.

---

> > ### Author Response · Authors · 2025-08-08
> >
> > We sincerely appreciate your positive assessment and feedback. Your insightful questions helped us demonstrate the BrainMoE for a multimodal (fMRI+EEG) scenario. We will revise the final version by the superior performance of combining fMRI and EEG experts, which will further strengthen our work and make BrainMoE more fundamental for brain modeling.
> >
> > We would be truly grateful if you could consider increasing the rating score accordingly.

---

### Official Review · Reviewer_hbEn · 2025-07-02

**Clarity:** 3
**Significance:** 3
**Originality:** 4
**Rating:** 5
**Confidence:** 4

**Summary:**

The manuscript proposes BrainMoE, a mixture-of-experts (MoE) framework for building brain foundation models by leveraging task-based as well as resting-state fMRI data across multiple cognitive states. Unlike prior models trained primarily on resting-state data, BrainMoE stratifies pretraining by cognitive states, training separate expert models for each of 12 distinct behavioral tasks. These expert embeddings are then fused via a cognition adapter and fine-tuned to support diverse downstream tasks (e.g., disease classification, behavior prediction, sex classification) across varied datasets and preprocessing pipelines. Empirical results across seven datasets demonstrate consistent and often substantial performance increase over BrainMass and BrainJEPA.

**Questions:**

1. The Methods suggest that the number of experts, N, equals the number of cognitive tasks. Why are there 36 experts in the end? There is a contradiction in the description.

2. Training data are primarily from UKB, an aging population. How does that impact results on ABIDE,  a primarily young population.

3. Do all three datasets have all 12 tasks?

4. In Table 1, HCPA and HCPYA are also considered downstream tasks. I thought those are tasks used for pre-training the model. Is it still fair to evaluate on the 3-task or 7-task classification as a downstream task?

**Ethical Concerns:**

["NO or VERY MINOR ethics concerns only"]

**Final Justification:**

The authors reasonably addressed my concerns. My original rating was already high. The novelty and results warrant publication.

**Limitations:**

In my experience, study different (or scanner difference) is a major hurdle for generalizability. The authors should acknowledge this limitation and also the lack of handling other confounding effects in the downstream classification.

**Quality:**

3

**Strengths And Weaknesses:**

Strengths:

* Incorporating task-fMRI in building foundation models and using MoE to resolve the heterogeneous connectivity patterns linked cognitive states from different tasks is a well justified reasonable technical proposal

* Experiments clearly indicate the performance gain from the MoE model.

Weaknesses:

* Writing of the Methods section can be improved. Although I understand the overall rationale, I'm still confused about Section 3.3 on how to select top-k experts. The confusion partly stems from the discrepancy between figure 3 and figure 4 that seemingly suggest different architectures of the model.

* The cognitive classification task seems to be a trivial task. I'm not sure how it can help self-supervised representation learning. Even when you train 12 separate encoders on exactly the same training data, you can freely get 12 versions of embeddings (e.g., just by applying a translation to the embeddings, you have a naively constructed different embeddings). Now the data for different experts are different, and the experts are trained independently, so I'm not sure why is there a need to further enforce a classification loss, a trivial task.

* Model interpretation is non-informative. Yes, some regions are picked, and with more experts, more regions are picked, but this is not something particularly interesting. I'd be interested to know whether each expert is linked to a specific cognitive construct as promised, and for each disease which expert (cognitive domain) is prioritized.

---

> ### Author Rebuttal · Authors · 2025-07-30
>
> ## **W1**
>
> We appreciate your suggestion, and we are committed to thoroughly revising the method section. Specifically, we will highlight the innovation of BrainMoE and connect Figs 3 and 4 with a smooth description as in this rebuttal: We illustrated the overall framework of BrainMoE in Fig. 3, including pre-training and fine-tuning, respectively. Fig. 4 illustrates the architecture of novel blocks shown in Fig. 3, particularly the brain expert and the cognition adapter. Figure 4 (a) shows the pre-training of brain experts corresponding to the individual expert in Figure 3 (a). On the other hand, Fig. 4 (b) shows the fine-tuning for downstream, and it corresponds to Fig. 3 (b), where the cognition adapter is the main block in the fine-tuning stage. We will combine Fig. 3 and Fig. 4 as one figure to clarify the method description.
>
> ## **W2**
>
> The motivation behind this is considering the variations of brain expert architecture to the extent of supervision type. The utilization of classification loss complements the gap between standard brain classification models, e.g., transformer-based [3][4], and the brain foundation models. Empirical evidence in our work shows that classification loss derived from standard brain modeling can benefit the performance of BrainMoE on clinical (Cog. Classif. in Table 3) and non-clinical applications (All-in-one in Table 4), indicating the potential of standard brain classification architectures for brain foundation modeling. We will clarify this motivation in the Introduction section and discuss the results in the Conclusion section.
>
> ## **W3**
>
> Here, we collect the expert routing probabilities of BrainMoE on clinical datasets and count the combination of top experts (top 1 if the probability > 0.9, otherwise top 2). We can observe the most frequent combination (Table C for reviewer PTHj):
>  - AD: [Working memory] 31.1%, [Rest, Motor] 20%
>  - ASD: [Language, Relational (relationships between stimuli)] 14.9%, [Rest, Relational] 8.3%
>  - PD: [Language, Relational] 20%, [Vismotor (color change and button press), Relational] 9.1%
>  - SZ: [Gambling, Relational] 18.5%, [Facename (face and name recall)] 14.3%
>
> The relatively more frequent activation of these expert combinations, e.g., [Language, Relational] for ASD and PD, suggests that BrainMoE captures complementary neural patterns that individual experts might miss.
>
> On the other hand, combinations for different applications align with neuroscientific knowledge:
>  - For AD, the dominance of [Working memory] aligns with that AD is commonly attributed to central executive impairment and frontal lobe dysfunction [1].
>  - For ASD, [Language, Relational] mirrors impairments in relational memory processing, where individuals with ASD show difficulties in forming relations between items and encoding relational but not item information [2], along with the language network showing atypical functional connectivity [3].
>  - For PD, [Vismotor, Relational] reflects the visuomotor processing deficits, including impaired decision-making cascades and altered cortical sensorimotor processing [4].
>  - For SZ, those combinations reflect deficits in reward learning and decision-making processes, as well as impaired facial processing and recognition abilities that are characteristic of the disorder [5].
>
> [1] 10.1016/j.cortex.2010.12.002
>
> [2] 10.1002/aur.1493
>
> [3] 10.1002/aur.2171
>
> [4] 10.3390/brainsci13081173
>
> [5] 10.3389/fneur.2019.00990
>
> ## **Q1**
>
> Sorry for the confusing description of the expert number. The number of experts (N) is derived from the number of distinct cognitive states of arbitrary input type (FC or BOLD) required by an arbitrary expert model architecture. Expert number is not strictly equal to the number of cognitive tasks but $N_{arch}\times N_{cog}$, where $N_{arch}$ is the number of available model architecture and $N_{cog}$ is the number of distinct cognitive states. In practice, there are three types of expert architecture (BrainMass, BrainJEPA, and Cog. Classif.) implemented in our BrainMoE All-in-one, so that $3\times 12=36$. We will revise the description in L.113-L.120 to clarify the expert number.
>
> ## **Q2**
>
> HCPYA has an age range 22 to 35 including young adults. But yes, given ABIDE has a younger population (age < 22), the performance of BrainMoE on ABIDE (ASD) is impacted as the worst among binary classifications (Table 1).
>
> ## **Q3**
>
> No. UKB has 2: Resting and task-Hariri. HCPA has 4: Resting, task-Carit, task-Facename, and task-Vismotor. HCPYA has 7: Motor, gambling, language, social, relational, emotion, and working memory. In total, there are 12 different cognitive states.
>
> ## **Q4**
>
> It is a fair comparison on HCPA and HCPYA datasets since the same cohorts of data were used for different models in the pre-training stage. Baselines and BrainMoE both have the same 68k pre-training data with the same number of training iterations. Although there are multiple experts in BrainMoE, each expert only sees a specific cognitive state, and each sample only has one cognitive state. Thus, BrainMoE has no additional pre-training epochs or iterations than baseline models, resulting in a fair comparison.
>
> ## **Lim.1**
>
> We appreciate your suggestion, and we are committed to thoroughly revising the ‘Limitations’ section with the discussion of confounding effects from study/scanner difference.

---

### Official Review · Reviewer_PTHj · 2025-07-03

**Clarity:** 2
**Significance:** 2
**Originality:** 3
**Rating:** 4
**Confidence:** 3

**Summary:**

This paper introduces a framework for fMRI understanding foundation model, called BrainMoE. It first trains each expert using fMRI data from corresponding datasets and then fine-tunes the fMRI representations by training a router and a query-Transformer. The authors conducted a number of experiments on a categorization task to evaluate their BrainMoE.

**Questions:**

After carefully reading the method section, I still have the following questions:

+ There should be significant differences among the datasets, such as variations in the dimensions of the BOLD signals. How does BrainMoE address this issue? I don’t seem to have found any explanation regarding this in the paper.

+ The authors introduced a binary supervision signal during pretraining to indicate whether the data is from resting-state fMRI. What is the benefit of this approach in the context of pretraining?

+ In Figure 3(b), the representations used for downstream tasks appear to come from a combination of multiple brain experts, whereas in Figure 4(a), the representations seem to be derived from any single model. This is somewhat unclear.

**Ethical Concerns:**

["NO or VERY MINOR ethics concerns only"]

**Final Justification:**

The authors’ response has addressed all of my concerns. Their explanations clearly demonstrate the novelty of the work. The training results on the decoding task indicate that BrainMoE is likely adaptable to this task. So, I am raising my rating.

**Limitations:**

I believe one of the main limitations of the paper lies in its experimental evaluation. BrainMoE is a robust fMRI representation model, and evaluating it solely on classification tasks is, in my opinion, insufficient. Assessing its performance on other types of tasks would significantly enhance the value of the paper — for example, tasks like visual decoding [1] (i.e., reconstructing visual stimuli [2] from fMRI signals).

The other minor limitation is the writing, figures, and formatting of the paper need further improvement. The expression in the introduction and methods sections requires additional polishing. The figures should be more aesthetically pleasing and concise, and the formatting needs further revision (for example, part of the text is obscured in Figure 3).

[1] Emily J. Allen et al. A massive 7T fMRI dataset to bridge cognitive neuroscience and artificial intelligence. Nature neuroscience.

[2] Paul S. Scotti et al. Mindeye2: Shared-subject models enable fmri-to-image with 1 hour of data. ICML 2024.

**Quality:**

3

**Strengths And Weaknesses:**

The strength of this paper lies in its massive experimental evaluation, which includes three pretraining datasets and seven downstream task datasets. However, its weaknesses include the lack of insight or novelty in the BrianMoE approach, the issue of homogeneity among the downstream tasks used for evaluation, and the overall clarity of the writing, which I believe needs further revision for better expression.

---

> ### Author Rebuttal · Authors · 2025-07-30
>
> ## Brief
>
> Thank you for your insightful comments and questions. The main concern of your reviews is derived from the misunderstanding of our methodological novelty and the lack of evaluations other than classification. We added results from a generic MoE to show the novelty of our BrainMoE approach, as well as regressive and generative applications.
>
> ## **W1**
>
> Along with the agreement of our novelty by reviewer Yxzq, one key novelty is innovating the BrainMoE approach that can significantly enhance the robustness of brain foundation model by pre-training with tasking-state fMRI, which was not fully explored in previous studies. Due to various sample sizes of pre-training datasets (Table A for Reviewer ZhRL), the data unbalance issues of MoE are amplified in the brain modeling area, as indicated by the performance of a generic MoE framework, the late fusion MoE [4], in Table A. To address this limitation, our BrainMoE presents innovative learning strategies for our new cognitive MoE adapter, to tackle the data unbalance issue.
>
> **Effectiveness of BrainMoE**: We can see late fusion MoE can sometimes outperform the baseline, but it is consistently worse than BrainMoE except for the smallest dataset Taowu (n=40). Unlike BrainMoE has the cognition adapter to implicitly utilize expert embeddings, the late fusion explicitly combines expert predictions, therefore cannot handle the unbalance issue.
>
> ---
> *Table A*
>
> ||ADNI|ABIDE|PPMI|Taowu|SZ|HCPA|HCPYA|
> |-|-|-|-|-|-|-|-|
> |Baseline|80.70$_{\pm7.85}$|67.81$_{\pm3.91}$|59.77$_{\pm14.22}$|68.00$_{\pm21.46}$|77.63$_{\pm8.56}$|90.63$_{\pm0.74}$|81.27$_{\pm1.27}$|
> |Late fusion MoE|73.33$_{\pm10.87}$|69.89$_{\pm306}$|61.11$_{\pm15.29}$|**91.24$_{\pm11.72}$**|76.95$_{\pm9.50}$|94.96$_{\pm2.64}$|88.58$_{\pm3.39}$|
> |BrainMoE|**81.23$_{\pm11.00}$**|**70.26$_{\pm3.40}$**|**62.97$_{\pm13.94}$**|88.57$_{\pm12.51}$|**83.10$_{\pm11.33}$**|**96.67$_{\pm0.77}$**|**93.19$_{\pm0.72}$**|
>
>
> Regarding top expert combinations (top 1 if the probability > 0.9, otherwise top 2) chosen by the router, it favors the top activated experts rather than the domain experts across all datasets.
>
> As listed in Table B, the combination from late fusion consistently contains Rest (n=29,971), Hariri (n=24,672), and Relational (n=1,609) experts. In contrast, BrainMoE has diverse and similar top combinations for heterogeneous and homogeneous applications, respectively (Table C). The contrast of selected combinatory patterns between routers trained by BrainMoE approach and the late fusion provides neuroscientific insight and supports our methodological novelty.
>
>
> ---
> *Table B*
>
> |(%)|ADNI|ABIDE|PPMI|Taowu|SZ|HCPA|HCPYA|
> |-|-|-|-|-|-|-|-|
> |Carit|0|0|0|0|0|5|0|
> |Emotion|0|4.6|1.4|0|0|2.8|0|
> |Emotion-Hariri|0|0|4.3|0|0|0|0|
> |Emotion-Motor|0|0|0|7.5|0|0|0|
> |Facename|18.5|4.4|0|0|0|2.8|0|
> |Facename-Relational|0|0|3.3|0|0|0|0|
> |Gambling|0|5.3|1|5|9.5|14|0|
> |Gambling-Motor|0|0|5.3|2.5|0|0|0|
> |Gambling-Relational|0|0|0.5|5|0|0.2|0|
> |Hariri|8.1|2.1|10.5|10|**47.1**|**16.3**|**27.1**|
> |Language|0|8.1|9.6|7.5|0|0|2.8|
> |Language-Relational|0|10.9|0|2.5|0|0|0|
> |Motor|0|0|0|7.5|0|4.8|4.4|
> |Motor-Hariri|0|0|0|5|0|0.2|0.5|
> |Relational|**20**|**15.2**|**22**|**25**|10.1|28|**22.6**|
> |Rest|**45.9**|10|**12.4**|0|**30.2**|2.5|20|
> |Rest-Emotion|0|0|5.7|0|0|0|0|
> |Rest-Hariri|5.9|0|4.8|0|0|0.1|1.6|
> |Rest-Motor|0|0|4.3|2.5|0|0|1.5|
> |Rest-Relational|0|**18.8**|5.7|0|0|0.3|0.5|
> |Rest-WM|0|0|0|5|0|0|0.2|
> |Social|0|0|0|2.5|0|5.3|0|
> |Social-Relational|0|0|0|5|0|0.2|0|
> |WM|0|8.2|0|2.5|0.5|**14.2**|7.9|
> |WM-Hariri|0|4.4|0|0|1.1|0|0.3|
> |WM-Motor|0|0|9.1|0|0|0|0|
> |WM-Relational|0|0|0|0|0|0.2|6.6|
>
>
> ---
> *Table C*
>
> |(%)|ADNI|ABIDE|PPMI|Taowu|SZ|HCPA|HCPYA|
> |-|-|-|-|-|-|-|-|
> |Carit-Facename|0.0|7.7|0.0|0.0|0.0|0.0|0.0|
> |Emotion-WM|0.0|0.0|3.8|10.0|0.0|0.0|0.4|
> |Facename|0.0|1.9|0.5|5.0|**14.3**|4.9|2.2|
> |Facename-Relational|0.0|0.0|1.9|2.5|0.0|4.4|0.1|
> |Gambling|7.4|3.9|0.5|5.0|0.0|2.2|**9.4**|
> |Gambling-Relational|0.0|5.9|0.0|0.0|**18.5**|0.0|3.6|
> |Hariri|0.0|0.0|2.9|0.0|3.7|10.7|**11.8**|
> |Language|0.7|3.9|5.3|**15.0**|0.0|**13.0**|9.0|
> |Language-Emotion|0.0|0.2|5.7|0.0|0.0|0.1|0.1|
> |Language-Facename|0.0|7.0|0.0|0.0|3.7|0.1|0.3|
> |Language-Motor|0.0|0.0|4.3|0.0|0.0|0.0|0.2|
> |Language-Relational|0.0|**14.9**|5.7|**20.0**|0.0|0.3|1.5|
> |Language-Rest|8.1|0.0|4.8|5.0|0.0|1.6|1.4|
> |Motor|0.0|0.2|0.0|0.0|0.0|11.5|2.0|
> |Motor-Relational|0.0|7.3|0.0|10.0|0.0|0.7|0.6|
> |Relational|0.0|3.9|4.3|12.5|14.3|**16.4**|7.3|
> |Rest|3.0|1.6|7.2|2.5|7.9|11.3|7.2|
> |Rest-Gambling|0.0|0.0|0.0|0.0|0.0|0.2|6.5|
> |Rest-Hariri|6.7|0.0|0.5|0.0|0.0|1.9|0.5|
> |Rest-Motor|**20.0**|0.0|1.0|0.0|5.8|0.0|0.1|
> |Rest-Relational|0.0|**8.3**|1.9|0.0|0.0|0.2|1.8|
> |Rest-Social|0.0|0.0|1.0|10.0|9.5|0.0|0.0|
> |Social|0.0|0.0|1.0|0.0|0.5|4.5|0.0|
> |Vismotor-Carit|0.0|6.1|0.0|0.0|0.0|0.0|0.0|
> |Vismotor-Relational|0.0|0.0|**9.1**|0.0|0.0|0.2|0.0|
> |Vismotor-Rest|0.0|0.0|0.0|0.0|6.3|0.1|0.0|
> |Vismotor-WM|8.9|0.0|0.0|0.0|0.0|0.0|0.0|
> |WM|**31.1**|3.3|0.0|0.0|0.5|2.2|6.7|
> |WM-Facename|0.0|6.7|0.0|0.0|0.0|0.0|0.0|
> |WM-Gambling|12.6|0.9|0.0|0.0|0.0|0.3|0.1|
> |WM-Hariri|0.0|0.0|7.2|0.0|6.3|0.0|1.2|
> |WM-Motor|0.0|0.4|**9.1**|0.0|0.0|0.0|0.1|
> |WM-Relational|0.0|6.3|0.0|0.0|3.7|1.3|3.4|
> |WM-Social|0.0|0.0|5.3|0.0|0.0|0.0|0.0|
>
> ## **W2**
>
> The challenge of heterogeneity from downstream tasks among our 7 datasets is as critical as previous brain foundation models (Appendix Table 6). The number of different downstream datasets is the highest, i.e., 5 cohorts, compared to the most recent brain foundation models. In addition, downstream tasks are also more variable than others since there are four types of brain disorders (AD, ASD, PD, SZ) and two types of non-clinical tasks (sex, cognitive state). We will add the variations of our downstream tasks to Table 6.
>
> ## **W3, Lim.2**
>
> We appreciate your suggestion, and we are committed to thoroughly revising the manuscript. Specifically, we will highlight the innovation of BrainMoE and provide more comprehensive supporting results as shown in this rebuttal.
>
> ## **Q1**
>
> BOLD signal is framed in a spatial-temporal data structure. One novelty of BrainMoE is modeling the brain with arbitrary input type (see L.68 point #2). Input of brain experts is denoted by **X** (L.106), which has variable temporal dimensions. Arbitrary brain expert maps **X** to **Z** (L.137) to have the same dimension $C_{hid}$ for BrainMoE modules. Although in practice, input shapes $C_{in}=M$ for BrainMass experts and $C_{in}=T$ for BrainJEPA experts are different, their cognitive embeddings both have the same shape $C_{hid}$. Therefore, BrainMoE can work with various input shapes. We will revise the method section with some examples of input shape to smooth the description of BrainMoE handling variable input shape.
>
> ## **Q2**
>
> The motivation behind this is considering the variations of brain expert architecture to the extent of supervision type. The utilization of binary supervision complements the gap between standard brain classification models, e.g., transformer-based [3], and the brain foundation models. Empirical evidence in our work shows that binary supervision can benefit the performance of BrainMoE on clinical (Cog. Classif. in Table 3) and non-clinical applications (All-in-one in Table 4), indicating the potential of standard brain classification architectures for brain foundation modeling. We will clarify this motivation in the Introduction section and discuss the results in the Conclusion section.
>
> ## **Q3**
>
> We illustrated the overall framework of BrainMoE and the learning strategy in Fig 3, including pre-training and fine-tuning. Fig 4 illustrates the novel blocks in our framework, particularly the brain expert and the cognition adapter. Fig 4a shows the pre-training of brain experts corresponding to Fig 3a instead of b, the downstream tasks are therefore not involved in this step. On the other hand, Fig 4b shows the fine-tuning for downstream, and it corresponds to Fig 3b. We will combine Fig 3 and Fig 4 as one figure to make the method description clear.
>
>
> ## **Lim.1**
>
> Age regression for 6 datasets has been evaluated for the best baseline, 68k version of BrainMass, and BrainMoE. We can see the performance of BrainMoE is the best. The improvement is especially significant for ABIDE (n=1,025, 6-58 yrs) with MSE 36.77 -> 4.86. This empirical evidence further supports the robustness of BrainMoE.
>
> ---
> *Table D*
>
> |MSE (unit: yr)|ADNI (age)|ABIDE (age)|PPMI (age)|Taowu (age)|HCPA (age)|HCPYA (age)|
> |-|-|-|-|-|-|-|
> |BrainMass-68k|36.28$_{\pm19.83}$|36.77$_{\pm17.21}$|33.09$_{\pm21.87}$|38.10$_{\pm28.33}$|22.66$_{\pm7.51}$|5.46$_{\pm2.69}$|
> |BrainMoE|**36.27$_{\pm 10.23}$**|**4.86$_{\pm2.67}$**|**29.89$_{\pm8.39}$**|**30.72$_{\pm12.25}$**|**10.56$_{\pm2.86}$**|**3.45$_{\pm1.08}$**|
>
> Visual decoding task for a new dataset NSD [1] has also been evaluated for MindEye2 [2] as the baseline and BrainMoE. We pre-trained two MindEye2s as the specific experts for long-term (novel trials) and short-term memory (easy/hard trials), respectively. Since visual decoding is a generative task, output contains much higher dimensions (256x1664 vs. class number 2-7) than downstream tasks focused on our study. Therefore, we skipped our cognition adapter by weighted summing the diffusion prior of two experts with the BrainMoE routing probabilities. Both baseline and BrainMoE are pretrained with subjects 2-7 and finetuned with subject 1 on the entire 40 sessions. The final train and test losses, cosine similarity, and Mean Squared Error (MSE) during finetuning are listed in Table E. Given the evidence that the performance of BrainMoE is better than the single expert MindEye2, there is potential for BrainMoE to expand to visual decoding.
>
> ---
> *Table E*
>
> ||MindEye2|BrainMoE|
> |-|-|-|
> |Train loss|9.639|**7.994**|
> |Test loss|11.142|**9.405**|
> |Cos. Sim.|0.778|**0.840**|
> |MSE|0.301|**0.261**|
>
> [1] 10.1038/s41593-021-00962-x
>
> [2] 10.48550/arXiv.2403.11207
>
> [3] 10.48550/arXiv.2210.06681
>
> [4] 10.48550/arXiv.2402.03226

---

### Official Review · Reviewer_ZhRL · 2025-07-04

**Clarity:** 3
**Significance:** 2
**Originality:** 2
**Rating:** 3
**Confidence:** 4

**Summary:**

This paper introduces a MoE-based foundation model for brain modeling. BrainMoE pre-trains expert models specific to different cognitive states, aggregates their embeddings using a Transformer-based cognition adapter, and demonstrates improved downstream performance on disease diagnosis, behavior, and sex classification across seven datasets.

**Questions:**

1. How is the number of experts chosen?
2. MoE can be prone to suffer from the unbalanced issue, i.e., only a few experts are activated and trained. Does BrainMoE has any regularization to address this?
3. Can you provide further analysis on which experts (cognitive tasks) are most relevant for specific downstream tasks, and whether these align with neuroscientific knowledge?

**Ethical Concerns:**

["NO or VERY MINOR ethics concerns only"]

**Final Justification:**

This paper is methodologically sound, with comprehensive experiments demonstrating significant performance gains. In the rebuttal phase, the authors adequately addressed most of my prior questions and concerns, including clarifications on the experiments and ablations.

However, the core idea of training independent experts for each cognitive state feels relatively straightforward and akin to a specialized ensemble method, while the cognition adapter primarily functions as a trainable classification head (via task-query embeddings) during fine-tuning. Overall, while the method is robust and well-executed, its novelty is somewhat limited, as it largely adapts existing Mixture-of-Experts concepts from other domains without groundbreaking innovations in the neuroimaging context.

Furthermore, given the relevance of cognitive states to diseases like Alzheimer's (AD), the paper could better relate individual experts to specific disease mechanisms and provide more in-depth neuroscientific analyses. For instance, while attention visualizations align with known brain networks (e.g., DMN for Autism, SMN for Parkinson's), deeper expert-level interpretations linking stratified cognitive embeddings to clinical outcomes would strengthen the contributions. This should be straightforward, as the main motivation of the paper is to develop separate experts for each cognitive state.

Given the above reasons, I would maintain my score as 3.

**Limitations:**

I did not see the discussion on limitations

**Quality:**

3

**Strengths And Weaknesses:**

Strength:
1. BrainMoE considers the task heterogeneity and pretrain dedicated experts for different cognitive states.
2. The approach is modular—any model pre-trained on a given cognitive state can be used as an expert. The cognition adapter further allows flexible downstream adaptation without restriction to a particular input type (BOLD or FC).
3. BrainMoE is tested across seven varied datasets from three different preprocessing pipelines, spanning clinical populations (e.g., Autism, Parkinson’s, Alzheimer's, Schizophrenia) and behavioral labeling, offeringh a comprehensive evaluation.

Weakness:
1. The results presented in Tables 1 and 2 are strong, but some standard baselines are omitted—for example, several datasets appear not to include a naive fine-tuning of a single (non-stratified) expert per cognitive state, or late fusion of independent predictions.
2. While Figure 5 highlights neurobiological alignment in attention maps, interpretability stops at the task-level. Analysis of which tasks or experts are favored for which downstream applications, and why certain combinations work better, would inform neuroscientific implications and practical use.
3. As discussed in Table 5, inference cost for BrainMoE is substantially higher than baselines (4–10x), which could limit large-scale or real-time applications, and there is limited discussion on possible amortized efficiency gains or pruning.

---

> ### Author Rebuttal · Authors · 2025-07-30
>
> ## Brief
>
> Thank you for your insightful comments and questions. The main concern of your reviews stands on the lack of evaluation per individual expert and late fusion. We have demonstrated all evaluations in our rebuttal. However, the evaluation per individual expert is lengthy and uncommon in previous brain modeling and MoE studies. We will keep them brief in the final version.
>
> ## **W1**
>
> Since resting-state fMRI was mainly used in previous brain foundation models (see L.40 in the manuscript), we included only resting-state experts as baselines. Here, we evaluated the standard baselines, a single expert per tasking-state, and reran the experiments of Tables 1 and 2 in the manuscript. Tables A and B show results for BrainMass-MLP baseline.
>
> **Baseline results** show that there are consistently existing task-specific experts (e.g., language for AD, working memory for PD) outperforming the resting-state baselines (30k version in the manuscript), confirming that task-state fMRI contains valuable information for brain modeling. Conclusively, BrainMoE still holds the best performance compared to all datasets and baselines. This supports that the utilization of cognitive embeddings from BrainMoE leads to more robustness of brain modeling than naively training a single task of fMRI. Furthermore, the expert combinations learned by BrainMoE and their strong relevance to downstream applications, shown in W2 response, such as Language for ASD, and Working memory for PD, offer valuable neuroscientific insights that current baseline models fail to capture.
>
> **The late fusion** of independent experts is also tested across 7 datasets. Please refer the results to Table A for Reviewer PTHj. The superior performance of BrainMoE and the contrast of selected expert routing patterns between BrainMoE and the late fusion provide neuroscientific insight and support our methodological novelty.
>
> ---
>  *Table A*
>
> |Expert|ADNI (AD)|ABIDE (ASD)|PPMI (PD)|Taowu (PD)|SZ (SZ)|HCPA (3-task)|HCPYA (7-task)|
> |-|-|-|-|-|-|-|-|
> |Hariri (n=24,672)|75.32$_{\\pm7.06}$|65.41$_{\\pm3.85}$|59.78$_{\\pm13.93}$|64.33$_{\\pm18.39}$|78.09$_{\\pm9.20}$|**89.42$_{\\pm0.64}$**|**77.58$_{\\pm2.35}$**|
> |Working memory (n=1,756)|76.15$_{\\pm6.37}$|61.31$_{\\pm3.71}$|**60.86$_{\\pm13.68}$**|**79.86$_{\\pm14.51}$**|78.34$_{\\pm9.30}$|80.53$_{\\pm0.53}$|64.59$_{\\pm1.05}$|
> |Motor (n=1,712)|76.35$_{\\pm6.26}$|60.30$_{\\pm4.37}$|57.86$_{\\pm11.09}$|72.57$_{\\pm26.98}$|**81.04$_{\\pm8.95}$**|78.74$_{\\pm0.71}$|60.41$_{\\pm1.93}$|
> |Gambling (n=1,705)|76.35$_{\\pm6.26}$|64.18$_{\\pm4.50}$|58.17$_{\\pm13.76}$|79.24$_{\\pm21.92}$|79.12$_{\\pm8.05}$|80.98$_{\\pm0.44}$|65.74$_{\\pm2.52}$|
> |Facename (n=1,699)|75.32$_{\\pm7.06}$|51.34$_{\\pm4.73}$|59.24$_{\\pm13.42}$|65.19$_{\\pm24.41}$|76.95$_{\\pm9.01}$|70.40$_{\\pm0.49}$|28.05$_{\\pm1.09}$|
> |Language (n=1,674)|**78.30$_{\\pm8.99}$**|**65.83$_{\\pm2.35}$**|60.84$_{\\pm14.54}$|64.43$_{\\pm28.40}$|80.63$_{\\pm8.30}$|78.73$_{\\pm0.69}$|58.08$_{\\pm1.00}$|
> |Social (n=1,674)|75.32$_{\\pm7.06}$|64.34$_{\\pm3.62}$|60.59$_{\\pm12.91}$|67.24$_{\\pm24.05}$|77.88$_{\\pm9.76}$|81.00$_{\\pm0.84}$|66.90$_{\\pm1.69}$|
> |Relational (n=1,609)|77.89$_{\\pm6.23}$|61.13$_{\\pm6.14}$|55.66$_{\\pm15.59}$|76.33$_{\\pm17.85}$|79.37$_{\\pm8.36}$|74.96$_{\\pm1.19}$|52.82$_{\\pm1.61}$|
> |Emotion (n=1,478)|76.52$_{\\pm6.18}$|60.29$_{\\pm3.83}$|58.46$_{\\pm15.24}$|69.67$_{\\pm22.97}$|78.01$_{\\pm8.71}$|85.01$_{\\pm0.90}$|70.69$_{\\pm1.55}$|
> |Carit (n= 698)|77.87$_{\\pm5.79}$|63.01$_{\\pm2.32}$|59.81$_{\\pm15.31}$|70.10$_{\\pm15.38}$|79.02$_{\\pm9.12}$|77.64$_{\\pm1.39}$|55.04$_{\\pm1.70}$|
> |Vismotor (n=637)|75.32$_{\\pm7.06}$|63.70$_{\\pm3.70}$|58.45$_{\\pm13.84}$|73.43$_{\\pm12.58}$|76.95$_{\\pm9.01}$|56.10$_{\\pm0.41}$|27.82$_{\\pm0.51}$|
>
> ---
>  *Table B*
>
> |Expert|ADNI (sex)|ABIDE (sex)|PPMI (sex)|Taowu (sex)|HCPA (sex)|HCPYA (sex)|
> |-|-|-|-|-|-|-|
> |Hariri (n=24,672)|56.25$_{\\pm9.06}$|75.13$_{\\pm4.97}$|**67.92$_{\\pm11.43}$**|67.86$_{\\pm27.70}$|**65.44$_{\\pm0.70}$**|**64.01$_{\\pm4.18}$**|
> |Working memory (n=1,756)|50.62$_{\\pm5.93}$|**75.92$_{\\pm4.43}$**|58.90$_{\\pm11.28}$|62.43$_{\\pm23.44}$|58.73$_{\\pm2.54}$|58.77$_{\\pm0.90}$|
> |Motor (n=1,712)|55.78$_{\\pm7.14}$|74.99$_{\\pm5.43}$|60.03$_{\\pm10.91}$|62.86$_{\\pm24.85}$|59.28$_{\\pm1.95}$|54.62$_{\\pm5.77}$|
> |Gambling (n=1,705)|61.17$_{\\pm7.01}$|74.76$_{\\pm5.56}$|63.37$_{\\pm8.16}$|70.00$_{\\pm28.67}$|63.52$_{\\pm1.05}$|61.22$_{\\pm1.12}$|
> |Facename (n=1,699)|65.61$_{\\pm7.95}$|73.87$_{\\pm5.66}$|54.65$_{\\pm7.45}$|52.10$_{\\pm26.55}$|58.36$_{\\pm2.47}$|59.69$_{\\pm2.97}$|
> |Language (n=1,674)|63.02$_{\\pm7.78}$|75.42$_{\\pm5.23}$|55.95$_{\\pm9.09}$|57.19$_{\\pm23.50}$|60.37$_{\\pm1.61}$|58.90$_{\\pm2.83}$|
> |Social (n=1,674)|60.09$_{\\pm5.19}$|74.55$_{\\pm5.04}$|62.87$_{\\pm10.80}$|69.05$_{\\pm26.14}$|64.30$_{\\pm1.55}$|59.91$_{\\pm2.28}$|
> |Relational (n=1,609)|**65.72$_{\\pm9.28}$**|75.67$_{\\pm6.28}$|55.51$_{\\pm9.74}$|**74.00$_{\\pm23.65}$**|57.84$_{\\pm1.72}$|57.32$_{\\pm3.33}$|
> |Emotion (n=1,478)|61.64$_{\\pm10.77}$|75.11$_{\\pm5.26}$|63.26$_{\\pm8.96}$|61.10$_{\\pm29.22}$|64.91$_{\\pm1.66}$|62.29$_{\\pm3.20}$|
> |Carit (n= 698)|56.31$_{\\pm17.07}$|75.36$_{\\pm4.96}$|63.23$_{\\pm9.37}$|58.86$_{\\pm24.63}$|59.75$_{\\pm1.06}$|59.23$_{\\pm2.81}$|
> |Vismotor (n=637)|63.91$_{\\pm5.26}$|73.87$_{\\pm5.66}$|64.04$_{\\pm8.85}$|62.10$_{\\pm23.25}$|46.42$_{\\pm1.76}$|46.65$_{\\pm5.00}$|
>
>
> ## **W2, Q3**
>
> Here, we collect the expert routing probabilities of BrainMoE on clinical datasets and count the combination of top experts (top 1 if the probability > 0.9, otherwise top 2). We can observe the most frequent combination (Table C for reviewer PTHj):
>  - AD: [Working memory] 31.1%, [Rest, Motor] 20%
>  - ASD: [Language, Relational (relationships between stimuli)] 14.9%, [Rest, Relational] 8.3%
>  - PD: [Language, Relational] 20%, [Vismotor (color change and button press), Relational] 9.1%
>  - SZ: [Gambling, Relational] 18.5%, [Facename (face and name recall)] 14.3%
>
> The relatively more frequent activation of these expert combinations, e.g., [Language, Relational] for ASD and PD, suggests that BrainMoE captures complementary neural patterns that individual experts might miss.
>
> On the other hand, combinations for different applications align with neuroscientific knowledge:
>  - For AD, the dominance of [Working memory] aligns with that AD is commonly attributed to central executive impairment and frontal lobe dysfunction [1].
>  - For ASD, [Language, Relational] mirrors impairments in relational memory processing, where individuals with ASD show difficulties in forming relations between items and encoding relational but not item information [2], along with the language network showing atypical functional connectivity [3].
>  - For PD, [Vismotor, Relational] reflects the visuomotor processing deficits, including impaired decision-making cascades and altered cortical sensorimotor processing [4].
>  - For SZ, those combinations reflect deficits in reward learning and decision-making processes, as well as impaired facial processing and recognition abilities that are characteristic of the disorder [5].
>
> [1] 10.1016/j.cortex.2010.12.002
>
> [2] 10.1002/aur.1493
>
> [3] 10.1002/aur.2171
>
> [4] 10.3390/brainsci13081173
>
> [5] 10.3389/fneur.2019.00990
>
> ## **W3**
>
> BrainMoE outperformed all baselines from the largest (n=4,863) to the smallest dataset (n=40), demonstrating the superior robustness. On the other hand, the running time of BrainMoE has fulfilled the real-time requirement for fMRI applications, which takes a much longer time (10-15mins) than BrainMoE (133-287ms) for data acquisition using a common 3T scanner [6].
>
> The possible amortized efficiency gains can be the pre-computation of cognitive embeddings. In fact, our experiments for BrainMass experts (FLOPs: 26.96M) have the cognitive embeddings computed on-the-fly, while BrainJEPA (FLOPs: 5.32T) computes them prior to the finetuning. Since the cognitive embeddings are frozen and fixed per data, pre-computation of the complex experts, BrainJEPA in our case, can save the time of iterative computation during BrainMoE finetuning. This trick enhanced efficiency as listed in Table 5, where BrainJEPA with the trillion-level FLOPs costs less time than BrainMass with the million level FLOPs. We will include the discussion of this possible amortized efficiency pruning in the “Limitation” section of the final version.
>
> [6] 10.3174/ajnr.A8067
>
> ## **Q1**
>
> The number of experts (N) equals $N_{arch}\times N_{cog}$, where $N_{arch}$ is the number of available model architectures and $N_{cog}$ is the number of distinct cognitive states. E.g., there are three types of architecture (BrainMass, BrainJEPA, and Cog. Classif.) implemented in our BrainMoE All-in-one, so that $3\times 12=36$. We will revise the description in L.113-L.120 to clarify the expert number.
>
> ## **Q2**
>
> Yes, the design of BrainMoE cognition adapter can address the unbalanced expert.
>
> As shown in Fig. 4 (b) and Eq. (3), the final prediction of BrainMoE is not the explicit fusion of expert embeddings but the transformation of input FC or BOLD weighted by the cross-attention between expert embeddings. This implicit usage of experts avoids their overfitted or underfitted cognition embeddings to directly bias the output of BrainMoE. Thus, it addressed unbalanced brain experts supported by empirical evidence:
>
> The pattern comparison (Table B vs. Table C for Reviewer PTHj) between the router trained by our BrainMoE approach and the generic late fusion MoE. Late fusion favors the top-activated experts rather than the domain experts across all datasets. As in Table B for Reviewer PTHj, the top combination among 7 datasets consistently contains Rest (n=29,971), Hariri (n=24,672), and Relational (n=1,609) experts. In contrast, as in Table C for Reviewer PTHj, BrainMoE has diverse top combinations for heterogeneous downstream tasks and similar top combinations for homogeneous applications, aligning with neuroscientific knowledge

---

> > ### Comment · Reviewer_ZhRL · 2025-08-01
> >
> > Thank the authors for their detailed response. My previous concerns are largely addressed. However, BrainMoE still exhibits a significant trade-off for its improved performance and computational cost. Although training individual experts for different cognitive states is interesting, this requires stratification of data and may not be easily extended. I have raised my score given the paper's merits.

---

> ### Author Response · Authors · 2025-08-01
>
> Thank you for your response and recognizing our work! Regarding to the last concern about the ease of stratification of data, the standard BIDS format [1] of neuroimaging data strictly requires the filename to have the cognitive state during imaging, i.e., BIDS format filename is like: {subject-name}\_{session-name}\_{task-name}\_{run-number}\_{more-info}.nii.gz. This makes the data stratification in our approach can be easily extended to other standard neuroimaing cohorts with BIDS format.
>
> [1] Gorgolewski, Krzysztof J., et al. "The brain imaging data structure, a format for organizing and describing outputs of neuroimaging experiments." Scientific data 3.1 (2016): 1-9.

---

### Decision · Program_Chairs · 2025-09-17

**Decision:**

Accept (poster)

**Comment:**

The manuscript introduces BrainMoE, a mixture-of-experts (MoE) framework for developing brain foundation models. BrainMoE leverages task-based and resting-state functional magnetic resonance imaging (fMRI) data across multiple cognitive states. Unlike prior models that were primarily trained on resting-state data, BrainMoE stratifies pretraining by cognitive state. It trains separate expert models for each distinct behavioral task. These expert embeddings are then fused via a cognition adapter and fine-tuned to support various downstream tasks, such as disease classification, behavior prediction, and sex classification, across several datasets and preprocessing pipelines. Empirical results across seven datasets demonstrate a consistent and often substantial increase in performance over current methods such as BrainMass and BrainJEPA.
Overall, the reviewers were pleased with the original idea and the extensive empirical validation. However, they also raised concerns about missing baselines, the lack of interpretability of the method, and inference cost.
Many of these concerns have been successfully addressed by the authors. Ultimately, the paper is sound, and the empirical results are interesting. However, while the method is robust and well-executed, it is not very novel, as it largely adapts existing Mixture-of-Experts concepts from other domains without groundbreaking innovations in the neuroimaging context. The paper does not provide significant neuroscientific insights.
Nevertheless, there is a consensus for acceptance.